# RNNLogic: Learning Logic Rules for Reasoning on Knowledge Graphs

**Meng Qu**[*1,2]**, Junkun Chen**[*3]**, Louis-Pascal Xhonneux**[1,2]**, Yoshua Bengio**[1,2,5]**, Jian Tang**[1,4,5]
[1]Mila - Québec AI Institute
[2]Université de Montréal
[3]Tsinghua University
[4]HEC Montréal
[5]Canadian Institute for Advanced Research (CIFAR)

## Abstract

This paper studies learning logic rules for reasoning on knowledge graphs. Logic rules provide interpretable explanations when used for prediction as well as being able to generalize to other tasks, and hence are critical to learn. Existing methods either suffer from the problem of searching in a large search space (e.g., neural logic programming) or ineffective optimization due to sparse rewards (e.g., techniques based on reinforcement learning). To address these limitations, this paper proposes a probabilistic model called RNNLogic. RNNLogic treats logic rules as a latent variable, and simultaneously trains a rule generator as well as a reasoning predictor with logic rules. We develop an EM-based algorithm for optimization. In each iteration, the reasoning predictor is first updated to explore some generated logic rules for reasoning. Then in the E-step, we select a set of high-quality rules from all generated rules with both the rule generator and reasoning predictor via posterior inference; and in the M-step, the rule generator is updated with the rules selected in the E-step. Experiments on four datasets prove the effectiveness of RNNLogic.

## 1 Introduction

Knowledge graphs are collections of real-world facts, which are useful in various applications. Each fact is typically specified as a triplet $(h, r, t)$ or equivalently $r(h, t)$, meaning entity $h$ has relation $r$ with entity $t$. For example, *Bill Gates* is the `Co-founder` of *Microsoft*. As it is impossible to collect all facts, knowledge graphs are incomplete. Therefore, a fundamental problem on knowledge graphs is to predict missing facts by reasoning with existing ones, a.k.a. knowledge graph reasoning.

This paper studies learning logic rules for reasoning on knowledge graphs. For example, one may extract a rule $\forall X, Y, Z \; \texttt{hobby}(X, Y) \leftarrow \texttt{friend}(X, Z) \wedge \texttt{hobby}(Z, Y)$, meaning that if $Z$ is a friend of $X$ and $Z$ has hobby $Y$, then $Y$ is also likely the hobby of $X$. Then the rule can be applied to infer new hobbies of people. Such logic rules are able to improve interpretability and precision of reasoning (Qu & Tang, 2019; Zhang et al., 2020). Moreover, logic rules can also be reused and generalized to other domains and data (Teru & Hamilton, 2020). However, due to the large search space, inferring high-quality logic rules for reasoning on knowledge graphs is a challenging task.

Indeed, a variety of methods have been proposed for learning logic rules from knowledge graphs. Most traditional methods such as path ranking (Lao & Cohen, 2010) and Markov logic networks (Richardson & Domingos, 2006) enumerate relational paths on graphs as candidate logic rules, and then learn a weight for each rule as an assessment of rule qualities. There are also some recent methods based on neural logic programming (Yang et al., 2017) and neural theorem provers (Rocktäschel & Riedel, 2017), which are able to learn logic rules and their weights simultaneously in a differentiable way. Though empirically effective for prediction, the search space of these methods is exponentially large, making it hard to identify high-quality logic rules. Besides, some recent efforts (Xiong et al., 2017) formulate the problem as a sequential decision making process, and use reinforcement learning to search for logic rules, which significantly reduces search complexity. However, due to the large action space and sparse reward in training, the performance of these methods is not yet satisfying.

---

[*]Equal contribution.

In this paper, we propose a principled probabilistic approach called RNNLogic which overcomes the above limitations. Our approach consists of a rule generator as well as a reasoning predictor with logic rules, which are simultaneously trained to enhance each other. The rule generator provides logic rules which are used by the reasoning predictor for reasoning, while the reasoning predictor provides effective reward to train the rule generator, which helps significantly reduce the search space. Specifically, for each query-answer pair, e.g., $q = (h, r, ?)$ and $a = t$, we model the probability of the answer conditioned on the query and existing knowledge graph $\mathcal{G}$, i.e., $p(a|\mathcal{G}, q)$, where a set of logic rules $z$ [1] is treated as a latent variable. The rule generator defines a prior distribution over logic rules for each query, i.e., $p(z|q)$, which is parameterized by a recurrent neural network. The reasoning predictor computes the likelihood of the answer conditioned on the logic rules and the existing knowledge graph $\mathcal{G}$, i.e., $p(a|\mathcal{G}, q, z)$. At each training iteration, we first sample a few logic rules from the rule generator, and further update the reasoning predictor to try out these rules for prediction. Then an EM algorithm (Neal & Hinton, 1998) is used to optimize the rule generator. In the E-step, a set of high-quality logic rules are selected from all the generated rules according to their posterior probabilities. In the M-step, the rule generator is updated to imitate the high-quality rules selected in the E-step. Extensive experimental results show that RNNLogic outperforms state-of-the-art methods for knowledge graph reasoning [2]. Besides, RNNLogic is able to generate high-quality logic rules.

## 2    RELATED WORK

Our work is related to existing efforts on learning logic rules for knowledge graph reasoning. Most traditional methods enumerate relational paths between query entities and answer entities as candidate logic rules, and further learn a scalar weight for each rule to assess the quality. Representative methods include Markov logic networks (Kok & Domingos, 2005; Richardson & Domingos, 2006; Khot et al., 2011), relational dependency networks (Neville & Jensen, 2007; Natarajan et al., 2010), rule mining algorithms (Galárraga et al., 2013; Meilicke et al., 2019), path ranking (Lao & Cohen, 2010; Lao et al., 2011) and probabilistic personalized page rank (ProPPR) algorithms (Wang et al., 2013; 2014a;b). Some recent methods extend the idea by simultaneously learning logic rules and the weights in a differentiable way, and most of them are based on neural logic programming (Rocktäschel & Riedel, 2017; Yang et al., 2017; Cohen et al., 2018; Sadeghian et al., 2019; Yang & Song, 2020) or neural theorem provers (Rocktäschel & Riedel, 2017; Minervini et al., 2020). These methods and our approach are similar in spirit, as they are all able to learn the weights of logic rules efficiently. However, these existing methods try to simultaneously learn logic rules and their weights, which is nontrivial in terms of optimization. The main innovation of our approach is to separate rule generation and rule weight learning by introducing a rule generator and a reasoning predictor respectively, which can mutually enhance each other. The rule generator generates a few high-quality logic rules, and the reasoning predictor only focuses on learning the weights of such high-quality rules, which significantly reduces the search space and leads to better reasoning results. Meanwhile, the reasoning predictor can in turn help identify some useful logic rules to improve the rule generator.

The other kind of rule learning method is based on reinforcement learning. The general idea is to train pathfinding agents, which search for reasoning paths in knowledge graphs to answer questions, and then extract logic rules from reasoning paths (Xiong et al., 2017; Chen et al., 2018; Das et al., 2018; Lin et al., 2018; Shen et al., 2018). However, training effective pathfinding agents is highly challenging, as the reward signal (i.e., whether a path ends at the correct answer) can be extremely sparse. Although some studies (Lin et al., 2018) try to get better reward by using embedding-based methods for reward shaping, the performance is still worse than most embedding-based methods. In our approach, the rule generator has a similar role to those pathfinding agents. The major difference is that we simultaneously train the rule generator and a reasoning predictor with logic rules, which mutually enhance each other. The reasoning predictor provides effective reward for training the rule generator, and the rule generator offers high-quality rules to improve the reasoning predictor.

Our work is also related to knowledge graph embedding, which solves knowledge graph reasoning by learning entity and relation embeddings in latent spaces (Bordes et al., 2013; Wang et al., 2014c; Yang et al., 2015; Nickel et al., 2016; Trouillon et al., 2016; Cai & Wang, 2018; Dettmers et al., 2018; Balazevic et al., 2019; Sun et al., 2019). With proper architectures, these methods are able to learn

---

[1]More precisely, $z$ is a multiset. In this paper, we use "set" to refer to "multiset" for conciseness.

[2]The codes of RNNLogic are available: `https://github.com/DeepGraphLearning/RNNLogic`

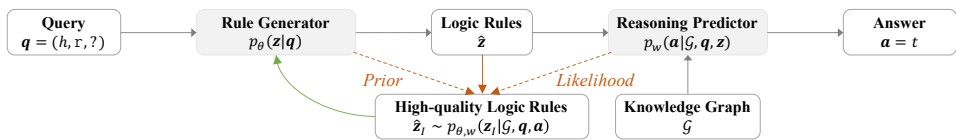

Figure 1: RNNLogic consists of a rule generator $p_\theta$ and a reasoning predictor $p_w$. Given a query, the rule generator generates logic rules for the reasoning predictor. The reasoning predictor takes the generated rules as input, and reasons on a knowledge graph to predict the answer. RNNLogic is optimized with an EM-based algorithm. In each iteration, the rule generator produces some logic rules, and we update the reasoning predictor to explore these rules for reasoning. Then in the E-step, a set of high-quality rules are identified from all generated rules via posterior inference. Finally in the M-step, the rule generator is updated to be consistent with the high-quality rules identified in E-step.

some simple logic rules. For example, TransE (Bordes et al., 2013) can learn some composition rules. RotatE (Sun et al., 2019) can mine some composition rules, symmetric rules and inverse rules. However, these methods can only find some simple rules in an implicit way. In contrast, our approach explicitly trains a rule generator, which is able to generate more complicated logic rules.

There are some works studying boosting rule-based models (Goldberg & Eckstein, 2010; Eckstein et al., 2017), where they dynamically add new rules according to the rule weights learned so far. These methods have been proven effective in binary classification and regression. Compared with them, our approach shares similar ideas, as we dynamically update the rule generator with the feedback from the reasoning predictor, but we focus on a different task, i.e., reasoning on knowledge graphs.

## 3 MODEL

In this section, we introduce the proposed approach RNNLogic which learns logic rules for knowledge graph reasoning. We first formally define knowledge graph reasoning and logic rules.

**Knowledge Graph Reasoning.** Let $p_{\text{data}}(\mathcal{G}, \boldsymbol{q}, \boldsymbol{a})$ be a training data distribution, where $\mathcal{G}$ is a background knowledge graph characterized by a set of $(h, \mathrm{r}, t)$-triplets which we may also write as $\mathrm{r}(h, t)$, $\boldsymbol{q} = (h, \mathrm{r}, ?)$ is a query, and $\boldsymbol{a} = t$ is the answer. Given $\mathcal{G}$ and the query $\boldsymbol{q}$, the goal is to predict the correct answer $\boldsymbol{a}$. More formally, we aim to model the probabilistic distribution $p(\boldsymbol{a}|\mathcal{G}, \boldsymbol{q})$.

**Logic Rule.** We perform knowledge graph reasoning by learning logic rules, where logic rules in this paper have the conjunctive form $\forall \{X_i\}_{i=0}^l \ \mathrm{r}(X_0, X_l) \leftarrow \mathrm{r}_1(X_0, X_1) \wedge \cdots \wedge \mathrm{r}_l(X_{l-1}, X_l)$ with $l$ being the rule length. This syntactic structure naturally captures composition, and can easily express other common logic rules such as symmetric or inverse rules. For example, let $\mathrm{r}^{-1}$ denote the inverse relation of relation $\mathrm{r}$, then each symmetric rule can be expressed as $\forall \{X, Y\} \ \mathrm{r}(X, Y) \leftarrow \mathrm{r}^{-1}(X, Y)$.

In RNNLogic, we treat a set of logic rules which could explain a query as a latent variable we have to infer. To do this, we introduce a rule generator and a reasoning predictor using logic rules. Given a query, the rule generator employs a recurrent neural network to generate a set of logic rules, which are given to the reasoning predictor for prediction. We optimize RNNLogic with an EM-based algorithm. In each iteration, we start with updating the reasoning predictor to try out some logic rules generated by the rule generator. Then in the E-step, we identify a set of high-quality rules from all generated rules via posterior inference, with the prior from the rule generator and likelihood from the reasoning predictor. Finally in the M-step, the rule generator is updated with the identified high-quality rules.

### 3.1 PROBABILISTIC FORMALIZATION

We start by formalizing knowledge graph reasoning in a probabilistic way, where a *set of logic rules $\boldsymbol{z}$* is treated as a latent variable. The target distribution $p(\boldsymbol{a}|\mathcal{G}, \boldsymbol{q})$ is jointly modeled by a rule generator and a reasoning predictor. The rule generator $p_\theta$ defines a prior over a set of latent rules $\boldsymbol{z}$ conditioned on a query $\boldsymbol{q}$, while the reasoning predictor $p_w$ gives the likelihood of the answer $\boldsymbol{a}$ conditioned on latent rules $\boldsymbol{z}$, the query $\boldsymbol{q}$, and the knowledge graph $\mathcal{G}$. Thus $p(\boldsymbol{a}|\mathcal{G}, \boldsymbol{q})$ can be computed as below:

$$p_{w,\theta}(\boldsymbol{a}|\mathcal{G}, \boldsymbol{q}) = \sum_{\boldsymbol{z}} p_w(\boldsymbol{a}|\mathcal{G}, \boldsymbol{q}, \boldsymbol{z}) p_\theta(\boldsymbol{z}|\boldsymbol{q}) = \mathbb{E}_{p_\theta(\boldsymbol{z}|\boldsymbol{q})}[p_w(\boldsymbol{a}|\mathcal{G}, \boldsymbol{q}, \boldsymbol{z})]. \tag{1}$$

The goal is to jointly train the rule generator and reasoning predictor to maximize the likelihood of training data. Formally, the objective function is presented as below:

$$\max_{\theta,w} \mathcal{O}(\theta, w) = \mathbb{E}_{(\mathcal{G},\boldsymbol{q},\boldsymbol{a}) \sim p_{\text{data}}}[\log p_{w,\theta}(\boldsymbol{a}|\mathcal{G}, \boldsymbol{q})] = \mathbb{E}_{(\mathcal{G},\boldsymbol{q},\boldsymbol{a}) \sim p_{\text{data}}}[\log \mathbb{E}_{p_\theta(\boldsymbol{z}|\boldsymbol{q})}[p_w(\boldsymbol{a}|\mathcal{G}, \boldsymbol{q}, \boldsymbol{z})]]. \quad (2)$$

## 3.2 Parameterization

**Rule Generator.** The rule generator defines the distribution $p_\theta(\boldsymbol{z}|\boldsymbol{q})$. For a query $\boldsymbol{q}$, the rule generator aims at generating a set of latent logic rules $\boldsymbol{z}$ for answering the query.

Formally, given a query $\boldsymbol{q} = (h, r, ?)$, we generate compositional logic rules by only considering the query relation $r$ without the query entity $h$, which allows the generated rules to generalize across entities. For each compositional rule in the abbreviation form $r \leftarrow r_1 \wedge \cdots \wedge r_l$, it can be viewed as a sequence of relations $[r, r_1, r_2 \cdots r_l, r_{\text{END}}]$, where $r$ is the query relation or the rule head, $\{r_i\}_{i=1}^l$ are the body of the rule, and $r_{\text{END}}$ is a special relation indicating the end of the relation sequence.

Such relation sequences can be effectively modeled by recurrent neural networks (Hochreiter & Schmidhuber, 1997), and thus we introduce $\text{RNN}_\theta$ to parameterize the rule generator. Given a query relation $r$, $\text{RNN}_\theta$ sequentially generates each relation in the body of a rule, until it reaches the ending relation $r_{\text{END}}$. In this process, the probabilities of generated rules are simultaneously computed. With such rule probabilities, we define the distribution over a set of rules $\boldsymbol{z}$ as a multinomial distribution:

$$p_\theta(\boldsymbol{z}|\boldsymbol{q}) = \text{Mu}(\boldsymbol{z}|N, \text{RNN}_\theta(\cdot|r)), \quad (3)$$

where Mu stands for multinomial distributions, $N$ is a hyperparameter for the size of the set $\boldsymbol{z}$, and $\text{RNN}_\theta(\cdot|r)$ defines a distribution over compositional rules with rule head being $r$. The generative process of a rule set $\boldsymbol{z}$ is quite intuitive, where we simply generate $N$ rules with $\text{RNN}_\theta$ to form $\boldsymbol{z}$.

**Reasoning Predictor with Logic Rules.** The reasoning predictor defines $p_w(\boldsymbol{a}|\mathcal{G}, \boldsymbol{q}, \boldsymbol{z})$. For a query $\boldsymbol{q}$, the predictor uses a set of rules $\boldsymbol{z}$ to reason on a knowledge graph $\mathcal{G}$ and predict the answer $\boldsymbol{a}$.

Following stochastic logic programming (Cussens, 2000), a principled reasoning framework, we use a log-linear model for reasoning. For each query $\boldsymbol{q} = (h, r, ?)$, a compositional rule is able to find different grounding paths on graph $\mathcal{G}$, leading to different candidate answers. For example, given query (*Alice*, hobby, ?), a rule hobby $\leftarrow$ friend $\wedge$ hobby can have two groundings, hobby(*Alice*, *Sing*) $\leftarrow$ friend(*Alice*, *Bob*) $\wedge$ hobby(*Bob*, *Sing*) and hobby(*Alice*, *Ski*) $\leftarrow$ friend(*Alice*, *Charlie*) $\wedge$ hobby(*Charlie*, *Ski*), yielding two candidate answers *Sing* and *Ski*.

Let $\mathcal{A}$ be the set of candidate answers which can be discovered by any logic rule in the set $\boldsymbol{z}$. For each candidate answer $e \in \mathcal{A}$, we compute the following scalar $\text{score}_w(e)$ for that candidate:

$$\text{score}_w(e) = \sum_{rule \in \boldsymbol{z}} \text{score}_w(e|rule) = \sum_{rule \in \boldsymbol{z}} \sum_{path \in \mathcal{P}(h, rule, e)} \psi_w(rule) \cdot \phi_w(path), \quad (4)$$

where $\mathcal{P}(h, rule, e)$ is the set of grounding paths which start at $h$ and end at $e$ following a *rule* (e.g., *Alice* $\xrightarrow{\text{friend}}$ *Bob* $\xrightarrow{\text{hobby}}$ *Sing*). $\psi_w(rule)$ and $\phi_w(path)$ are scalar weights of each *rule* and *path*. Intuitively, the score of each candidate answer $e$ is the sum of scores contributed by each rule, i.e., $\text{score}_w(e|rule)$. To get $\text{score}_w(e|rule)$, we sum over every grounding *path* found in the graph $\mathcal{G}$.

For the scalar weight $\psi_w(rule)$ of each *rule*, it is a learnable parameter to optimize. For the score $\phi_w(path)$ of each specific *path*, we explore two methods for parameterization. One method always sets $\phi_w(path) = 1$. However, this method cannot distinguish between different relational paths. To address the problem, for the other method, we follow an embedding algorithm RotatE (Sun et al., 2019) to introduce an embedding for each entity and model each relation as a rotation operator on entity embeddings. Then for each grounding *path* of *rule* starting from $h$ to $e$, if we rotate the embedding of $h$ according to the rotation operators defined by the body relations of *rule*, we should expect to obtain an embedding close to the embedding of $e$. Thus we compute the similarity between the derived embedding and the embedding of $e$ as $\phi_w(path)$, which can be viewed as a measure of the soundness and consistency of each *path*. For example, given a path *Alice* $\xrightarrow{\text{friend}}$ *Bob* $\xrightarrow{\text{hobby}}$ *Sing*, we rotate *Alice*'s embedding with the operators defined by friend and hobby. Afterwards we compute the similarity of the derived embedding and embedding of *Sing* as $\phi_w(path)$. Such a method allows us to compute a specific $\phi_w(path)$ for each *path*, which further leads to more precise scores for different candidate answers. See App. C for more details of the parameterization method.

Once we have the score for every candidate answer, we can further define the probability that the answer $\boldsymbol{a}$ of the query $\boldsymbol{q}$ is entity $e$ by using a softmax function as follows:

$$p_w(\boldsymbol{a} = e|\mathcal{G}, \boldsymbol{q}, \boldsymbol{z}) = \frac{\exp(\texttt{score}_w(e))}{\sum_{e' \in \mathcal{A}} \exp(\texttt{score}_w(e'))}. \tag{5}$$

### 3.3 OPTIMIZATION

Next, we introduce how we optimize the reasoning predictor and rule generator to maximize the objective in Eq. (2). At each training iteration, we first update the reasoning predictor $p_w$ according to some rules generated by the generator, and then update the rule generator $p_\theta$ with an EM algorithm. In the E-step, a set of high-quality rules are identified from all generated rules via posterior inference, with the rule generator as the prior and the reasoning predictor as the likelihood. In the M-step, the rule generator is then updated to be consistent with the high-quality rules selected in the E-step.

Formally, at each training iteration, we start with maximizing the objective $\mathcal{O}(\theta, w)$ in Eq. (2) with respect to the reasoning predictor $p_w$. To do that, we notice that there is an expectation operation with respect to $p_\theta(\boldsymbol{z}|\boldsymbol{q})$ for each training instance $(\mathcal{G}, \boldsymbol{q}, \boldsymbol{a})$. By drawing a sample $\hat{\boldsymbol{z}} \sim p_\theta(\boldsymbol{z}|\boldsymbol{q})$ for query $\boldsymbol{q}$, we can approximate the objective function of $w$ at each training instance $(\mathcal{G}, \boldsymbol{q}, \boldsymbol{a})$ as below:

$$\mathcal{O}_{(\mathcal{G}, \boldsymbol{q}, \boldsymbol{a})}(w) = \log \mathbb{E}_{p_\theta(\boldsymbol{z}|\boldsymbol{q})}[p_w(\boldsymbol{a}|\mathcal{G}, \boldsymbol{q}, \boldsymbol{z})] \approx \log p_w(\boldsymbol{a}|\mathcal{G}, \boldsymbol{q}, \hat{\boldsymbol{z}}) \tag{6}$$

Basically, we sample a set of rules $\hat{\boldsymbol{z}}$ from the generator and feed $\hat{\boldsymbol{z}}$ into the reasoning predictor. Then the parameter $w$ of the reasoning predictor is updated to maximize the log-likelihood of the answer $\boldsymbol{a}$.

With the updated reasoning predictor, we then update the rule generator $p_\theta$ to maximize the objective $\mathcal{O}(\theta, w)$. In general, this can be achieved by REINFORCE (Williams, 1992) or the reparameterization trick (Jang et al., 2017; Maddison et al., 2017), but they are less effective in our problem due to the large number of logic rules. Therefore, an EM framework is developed to optimize the rule generator.

**E-step.** Recall that when optimizing the reasoning predictor, we draw a set of rules $\hat{\boldsymbol{z}}$ for each data instance $(\mathcal{G}, \boldsymbol{q}, \boldsymbol{a})$, and let the reasoning predictor use $\hat{\boldsymbol{z}}$ to predict $\boldsymbol{a}$. For each data instance, the E-step aims to identify a set of $K$ high-quality rules $\boldsymbol{z}_I$ from all generated rules $\hat{\boldsymbol{z}}$, i.e., $\boldsymbol{z}_I \subset \hat{\boldsymbol{z}}, |\boldsymbol{z}_I| = K$.

Formally, this is achieved by considering the posterior probability of each subset of logic rules $\boldsymbol{z}_I$, i.e., $p_{\theta,w}(\boldsymbol{z}_I|\mathcal{G}, \boldsymbol{q}, \boldsymbol{a}) \propto p_w(\boldsymbol{a}|\mathcal{G}, \boldsymbol{q}, \boldsymbol{z}_I)p_\theta(\boldsymbol{z}_I|\boldsymbol{q})$, with prior of $\boldsymbol{z}_I$ from the rule generator $p_\theta$ and likelihood from the reasoning predictor $p_w$. The posterior combines knowledge from both the rule generator and reasoning predictor, so the likely set of high-quality rules can be obtained by sampling from the posterior. However, sampling from the posterior is nontrivial due to its intractable partition function, so we approximate the posterior using a more tractable form with the proposition below:

**Proposition 1** *Consider a data instance $(\mathcal{G}, \boldsymbol{q}, \boldsymbol{a})$ with $\boldsymbol{q} = (h, r, ?)$ and $\boldsymbol{a} = t$. For a set of rules $\hat{\boldsymbol{z}}$ generated by the rule generator $p_\theta$, we can compute the following score $H$ for each rule $\in \hat{\boldsymbol{z}}$:*

$$H(rule) = \left\{ \texttt{score}_w(t|rule) - \frac{1}{|\mathcal{A}|} \sum_{e \in \mathcal{A}} \texttt{score}_w(e|rule) \right\} + \log \texttt{RNN}_\theta(rule|r), \tag{7}$$

*where $\mathcal{A}$ is the set of all candidate answers discovered by rules in $\hat{\boldsymbol{z}}$, $\texttt{score}_w(e|rule)$ is the score that each rule contributes to entity $e$ as defined by Eq. (4), $\texttt{RNN}_\theta(rule|r)$ is the prior probability of rule computed by the generator. Suppose $s = \max_{e \in \mathcal{A}} |\texttt{score}_w(e)| < 1$. Then for a subset of rules $\boldsymbol{z}_I \subset \hat{\boldsymbol{z}}$ with $|\boldsymbol{z}_I| = K$, the log-probability $\log p_{\theta,w}(\boldsymbol{z}_I|\mathcal{G}, \boldsymbol{q}, \boldsymbol{a})$ can be approximated as follows:*

$$\left| \log p_{\theta,w}(\boldsymbol{z}_I|\mathcal{G}, \boldsymbol{q}, \boldsymbol{a}) - \left( \sum_{rule \in \boldsymbol{z}_I} H(rule) + \gamma(\boldsymbol{z}_I) + \text{const} \right) \right| \leq s^2 + O(s^4) \tag{8}$$

*where* const *is a constant term that is independent from $\boldsymbol{z}_I$, $\gamma(\boldsymbol{z}_I) = \log(K!/\prod_{rule \in \hat{\boldsymbol{z}}} n_{rule}!)$, with $K$ being the given size of set $\boldsymbol{z}_I$ and $n_{rule}$ being the number of times each rule appears in $\boldsymbol{z}_I$.*

We prove the proposition in App. A.1. In practice, we can apply weight decay to the weight of logic rules in Eq. (4), and thereby reduce $s = \max_{e \in \mathcal{A}} |\texttt{score}_w(e)|$ to get a more precise approximation.

The above proposition allows us to utilize $(\sum_{rule \in \boldsymbol{z}_I} H(rule) + \gamma(\boldsymbol{z}_I) + \text{const})$ to approximate the log-posterior $\log p_{\theta,w}(\boldsymbol{z}_I|\mathcal{G}, \boldsymbol{q}, \boldsymbol{a})$, yielding a distribution $q(\boldsymbol{z}_I) \propto \exp(\sum_{rule \in \boldsymbol{z}_I} H(rule) + \gamma(\boldsymbol{z}_I))$ as a

good approximation of the posterior. It turns out that the derived $q(\boldsymbol{z}_I)$ is a multinomial distribution, and thus sampling from $q(\boldsymbol{z}_I)$ is more tractable. Specifically, a sample $\hat{\boldsymbol{z}}_I$ from $q(\boldsymbol{z}_I)$ can be formed with $K$ logic rules which are independently sampled from $\hat{\boldsymbol{z}}$, where the probability of sampling each *rule* is computed as $\exp(H(\textit{rule}))/(\sum_{\textit{rule}'\in\hat{\boldsymbol{z}}}\exp(H(\textit{rule}')))$. We provide the proof in the App. A.2.

Intuitively, we could view $H(\textit{rule})$ of each *rule* as an assessment of the rule quality, which considers two factors. The first factor is based on the reasoning predictor $p_w$, and it is computed as the score that a *rule* contributes to the correct answer $t$ minus the mean score that this *rule* contributes to other candidate answers. If a rule gives higher score to the true answer and lower score to other candidate answers, then the rule is more likely to be important. The second factor is based on the rule generator $p_\theta$, where we compute the prior probability for each *rule* and use the probability for regularization.

Empirically, we find that picking $K$ rules with highest $H(\textit{rule})$ to form $\hat{\boldsymbol{z}}_I$ works better than sampling from the posterior. Similar observations have been made on the node classification task (Qu et al., 2019). In fact, $\hat{\boldsymbol{z}}_I$ formed by the top-$K$ rules is an MAP estimation of the posterior, and thus the variant of picking top-$K$ rules yields a hard-assignment EM algorithm (Koller & Friedman, 2009). Despite the reduced theoretical guarantees, we use this variant in practice for its good performance.

**M-step.** Once we obtain a set of high-quality logic rules $\hat{\boldsymbol{z}}_I$ for each data instance $(\mathcal{G}, \boldsymbol{q}, \boldsymbol{a})$ in the E-step, we further leverage those rules to update the parameters $\theta$ of the rule generator in the M-step.

Specifically, for each data instance $(\mathcal{G}, \boldsymbol{q}, \boldsymbol{a})$, we treat the corresponding rule set $\hat{\boldsymbol{z}}_I$ as part of the (now complete) training data, and update the rule generator by maximizing the log-likelihood of $\hat{\boldsymbol{z}}_I$:

$$\mathcal{O}_{(\mathcal{G},\boldsymbol{q},\boldsymbol{a})}(\theta) = \log p_\theta(\hat{\boldsymbol{z}}_I|\boldsymbol{q}) = \sum_{\textit{rule}\in\hat{\boldsymbol{z}}_I}\log\mathrm{RNN}_\theta(\textit{rule}|r) + \mathrm{const.} \tag{9}$$

With the above objective, the feedback from the reasoning predictor can be effectively distilled into the rule generator. In this way, the rule generator will learn to only generate high-quality rules for the reasoning predictor to explore, which reduces the search space and yields better empirical results.

For more detailed analysis of the EM algorithm for optimizing $\mathcal{O}(\theta, w)$, please refer to App. B.

---

**Algorithm 1** Workflow of RNNLogic

**while** not converge **do**
    For each instance, use the rule generator $p_\theta$ to generate a set of rules $\hat{\boldsymbol{z}}$ ($|\hat{\boldsymbol{z}}| = N$).
    For each instance, update the reasoning predictor $p_w$ based on generated rules $\hat{\boldsymbol{z}}$ and Eq. (6).
    ⊡ *E-step:*
    For each instance, identify $K$ high-quality rules $\hat{\boldsymbol{z}}_I$ from $\hat{\boldsymbol{z}}$ according to $H(\textit{rule})$ in Eq. (7).
    ⊡ *M-step:*
    For each instance, update the rule generator $p_\theta$ according to the identified rules and Eq. (9).
**end while**
During testing, for each query, use $p_\theta$ to generate $N$ rules and feed them into $p_w$ for prediction.

---

## 4 EXPERIMENT

### 4.1 EXPERIMENT SETTINGS

**Datasets.** We choose four datasets for evaluation, including FB15k-237 (Toutanova & Chen, 2015), WN18RR (Dettmers et al., 2018), Kinship and UMLS (Kok & Domingos, 2007). For Kinship and UMLS, there are no standard data splits, so we randomly sample 30% of all the triplets for training, 20% for validation, and the rest 50% for testing. The detailed statistics are summarized in the App. D.

**Compared Algorithms.** We compare the following algorithms in experiment:
⊡ *Rule learning methods.* For traditional statistical relational learning methods, we choose Markov logic networks (Richardson & Domingos, 2006), boosted relational dependency networks (Natarajan et al., 2010) and path ranking (Lao & Cohen, 2010). We also consider neural logic programming methods, including NeuralLP (Yang et al., 2017), DRUM (Sadeghian et al., 2019) and NLIL (Yang & Song, 2020). In addition, we compare against CTP (Minervini et al., 2020), a differentiable method based on neural theorem provers. Besides, we consider three reinforcement learning methods, which

Table 1: Results of reasoning on FB15k-237 and WN18RR. H@$k$ is in %. [*] means the numbers are taken from the original papers. [†] means we rerun the methods with the same evaluation process.

| Category | Algorithm | FB15k-237 | | | | | WN18RR | | | | |
|---|---|---|---|---|---|---|---|---|---|---|---|
| | | MR | MRR | H@1 | H@3 | H@10 | MR | MRR | H@1 | H@3 | H@10 |
| No Rule Learning | TransE* | 357 | 0.294 | - | - | 46.5 | 3384 | 0.226 | - | - | 50.1 |
| | DistMult* | 254 | 0.241 | 15.5 | 26.3 | 41.9 | 5110 | 0.43 | 39 | 44 | 49 |
| | ComplEx* | 339 | 0.247 | 15.8 | 27.5 | 42.8 | 5261 | 0.44 | 41 | 46 | 51 |
| | ComplEx-N3* | - | **0.37** | - | - | **56** | - | **0.48** | - | - | **57** |
| | ConvE* | 244 | 0.325 | 23.7 | 35.6 | 50.1 | 4187 | 0.43 | 40 | 44 | 52 |
| | TuckER* | - | 0.358 | **26.6** | **39.4** | 54.4 | - | 0.470 | 44.3 | 48.2 | 52.6 |
| | RotatE* | **177** | 0.338 | 24.1 | 37.5 | 53.3 | **3340** | 0.476 | 42.8 | 49.2 | **57.1** |
| Rule Learning | PathRank | - | 0.087 | 7.4 | 9.2 | 11.2 | - | 0.189 | 17.1 | 20.0 | 22.5 |
| | NeuralLP† | - | 0.237 | 17.3 | 25.9 | 36.1 | - | 0.381 | 36.8 | 38.6 | 40.8 |
| | DRUM† | - | 0.238 | 17.4 | 26.1 | 36.4 | - | 0.382 | 36.9 | 38.8 | 41.0 |
| | NLIL* | - | 0.25 | - | - | 32.4 | - | - | - | - | - |
| | M-Walk* | - | 0.232 | 16.5 | 24.3 | - | - | 0.437 | 41.4 | 44.5 | - |
| RNNLogic | w/o emb. | 538 | 0.288 | 20.8 | 31.5 | 44.5 | 7527 | 0.455 | 41.4 | 47.5 | 53.1 |
| | with emb. | 232 | 0.344 | 25.2 | 38.0 | 53.0 | 4615 | **0.483** | **44.6** | **49.7** | 55.8 |

Table 2: Results of knowledge graph reasoning on the FB15k-237 and WN18RR datasets with only $(h, \mathrm{r}, ?)$-queries. H@$k$ is in %. [*] means that the numbers are taken from the original papers.

| Category | Algorithm | FB15k-237 | | | | | WN18RR | | | | |
|---|---|---|---|---|---|---|---|---|---|---|---|
| | | MR | MRR | H@1 | H@3 | H@10 | MR | MRR | H@1 | H@3 | H@10 |
| Rule Learning | MINERVA* | - | 0.293 | 21.7 | 32.9 | 45.6 | - | 0.448 | 41.3 | 45.6 | 51.3 |
| | MultiHopKG* | - | 0.407 | 32.7 | - | 56.4 | - | 0.472 | 43.7 | - | 54.2 |
| RNNLogic | w/o emb. | 459.0 | 0.377 | 28.9 | 41.2 | 54.9 | 7662.8 | 0.478 | 43.8 | 50.3 | 55.3 |
| | with emb. | **146.1** | **0.443** | **34.4** | **48.9** | **64.0** | **3767.0** | **0.506** | **46.3** | **52.3** | **59.2** |

are MINERVA (Das et al., 2018), MultiHopKG (Lin et al., 2018) and M-Walk (Shen et al., 2018). ⊡ *Other methods.* We also compare with some embedding methods, including TransE (Bordes et al., 2013), DistMult (Yang et al., 2015), ComplEx (Trouillon et al., 2016), ComplEx-N3 (Lacroix et al., 2018), ConvE (Dettmers et al., 2018), TuckER (Balazevic et al., 2019) and RotatE (Sun et al., 2019). ⊡ *RNNLogic.* For RNNLogic, we consider two model variants. The first variant assigns a constant score to different grounding paths in the reasoning predictor, i.e., $\phi_w(path) = 1$ in Eq. (4), and we denote this variant as *w/o emb.*. The second variant leverages entity embeddings and relation embeddings to compute the path score $\phi_w(path)$, and we denote the variant as *with emb.*.

**Evaluation Metrics.** During evaluation, for each test triplet $(h, \mathrm{r}, t)$, we build two queries $(h, \mathrm{r}, ?)$ and $(t, \mathrm{r}^{-1}, ?)$ with answers $t$ and $h$. For each query, we compute a probability for each entity, and compute the rank of the correct answer. Given the ranks from all queries, we report the Mean Rank (MR), Mean Reciprocal Rank (MRR) and Hit@$k$ (H@$k$) under the filtered setting (Bordes et al., 2013), which is used by most existing studies. Note that there can be a case where an algorithm assigns the same probability to the correct answer and a few other entities. For such a case, many methods compute the rank of the correct answer as $(m + 1)$ where $m$ is the number of entities receiving higher probabilities than the correct answer. This setup can be problematic according to Sun et al. (2020). For fair comparison, in that case we compute the expectation of each evaluation metric over all the random shuffles of entities which receive the same probability as the correct answer. For example, if there are $n$ entities which have the same probability as the correct answer in the above case, then we treat the rank of the correct answer as $(m + (n + 1)/2)$ when computing Mean Rank.

Besides, we notice that in MINERVA (Das et al., 2018) and MultiHopKG (Lin et al., 2018), they only consider queries in the form of $(h, \mathrm{r}, ?)$, which is different from our default setting. To make fair comparison with these methods, we also apply RNNLogic to this setting and report the performance.

**Experimental Setup of RNNLogic.** For each training triplet $(h, \mathrm{r}, t)$, we add an inverse triplet $(t, \mathrm{r}^{-1}, h)$ into the training set, yielding an augmented set of training triplets $\mathcal{T}$. We use a closed-world assumption for model training, which assumes that any triplet outside $\mathcal{T}$ is incorrect. To build a training instance from $p_{\text{data}}$, we first randomly sample a triplet $(h, \mathrm{r}, t)$ from $\mathcal{T}$, and then form an instance as $(\mathcal{G} = \mathcal{T} \setminus \{(h, \mathrm{r}, t)\}, \boldsymbol{q} = (h, \mathrm{r}, ?), \boldsymbol{a} = t)$. Basically, we use the sampled triplet $(h, \mathrm{r}, t)$ to construct the query and answer, and use the rest of triplets in $\mathcal{T}$ to form the background knowledge graph $\mathcal{G}$. During testing, the background knowledge graph $\mathcal{G}$ is formed with all the triplets in $\mathcal{T}$. For the rule generator, the maximum length of generated rules is set to 4 for FB15k-237, 5 for WN18RR, and 3 for the rest, which are selected on validation data. See App. D for the detailed setting.

Table 3: Results of reasoning on the Kinship and UMLS datasets. H@$k$ is in %.

| Category | Algorithm | Kinship | | | | | UMLS | | | | |
|---|---|---|---|---|---|---|---|---|---|---|---|
| | | MR | MRR | H@1 | H@3 | H@10 | MR | MRR | H@1 | H@3 | H@10 |
| **No Rule Learning** | DistMult | 8.5 | 0.354 | 18.9 | 40.0 | 75.5 | 14.6 | 0.391 | 25.6 | 44.5 | 66.9 |
| | ComplEx | 7.8 | 0.418 | 24.2 | 49.9 | 81.2 | 13.6 | 0.411 | 27.3 | 46.8 | 70.0 |
| | ComplEx-N3 | - | 0.605 | 43.7 | 71.0 | 92.1 | - | 0.791 | 68.9 | 87.3 | 95.7 |
| | TuckER | 6.2 | 0.603 | 46.2 | 69.8 | 86.3 | 5.7 | 0.732 | 62.5 | 81.2 | 90.9 |
| | RotatE | 3.7 | 0.651 | 50.4 | 75.5 | 93.2 | 4.0 | 0.744 | 63.6 | 82.2 | 93.9 |
| **Rule Learning** | MLN | 10.0 | 0.351 | 18.9 | 40.8 | 70.7 | 7.6 | 0.688 | 58.7 | 75.5 | 86.9 |
| | Boosted RDN | 25.2 | 0.469 | 39.5 | 52.0 | 56.7 | 54.8 | 0.227 | 14.7 | 25.6 | 37.6 |
| | PathRank | - | 0.369 | 27.2 | 41.6 | 67.3 | - | 0.197 | 14.8 | 21.4 | 25.2 |
| | NeuralLP | 16.9 | 0.302 | 16.7 | 33.9 | 59.6 | 10.3 | 0.483 | 33.2 | 56.3 | 77.5 |
| | DRUM | 11.6 | 0.334 | 18.3 | 37.8 | 67.5 | 8.4 | 0.548 | 35.8 | 69.9 | 85.4 |
| | MINERVA | - | 0.401 | 23.5 | 46.7 | 76.6 | - | 0.564 | 42.6 | 65.8 | 81.4 |
| | CTP | - | 0.335 | 17.7 | 37.6 | 70.3 | - | 0.404 | 28.8 | 43.0 | 67.4 |
| **RNNLogic** | w/o emb. | 3.9 | 0.639 | 49.5 | 73.1 | 92.4 | 5.3 | 0.745 | 63.0 | 83.3 | 92.4 |
| | with emb. | **3.1** | **0.722** | **59.8** | **81.4** | **94.9** | **3.1** | **0.842** | **77.2** | **89.1** | **96.5** |

## 4.2 Results

**Comparison against Existing Methods.** We present the results on the FB15k-237 and WN18RR datasets in Tab. 1 and Tab. 2. The results on the Kinship and UMLS datasets are shown in Tab. 3.

We first compare RNNLogic with rule learning methods. RNNLogic achieves much better results than statistical relational learning methods (MLN, Boosted RDN, PathRank) and neural differentiable methods (NeuralLP, DRUM, NLIL, CTP). This is because the rule generator and reasoning predictor of RNNLogic can collaborate with each other to reduce search space and learn better rules. RNNLogic also outperforms reinforcement learning methods (MINERVA, MultiHopKG, M-Walk). The reason is that RNNLogic is optimized with an EM-based framework, in which the reasoning predictor provides more useful feedback to the rule generator, and thus addresses the challenge of sparse reward.

We then compare RNNLogic against state-of-the-art embedding-based methods. For RNNLogic with embeddings in the reasoning predictor (*with emb.*), it outperforms most compared methods in most cases, and the reason is that RNNLogic is able to use logic rules to enhance reasoning performance. For RNNLogic without embedding (*w/o emb.*), it achieves comparable results to embedding-based methods, especially on WN18RR, Kinship and UMLS where the training triplets are quite limited.

**Quality of Learned Logic Rules.** Next, we study the quality of rules learned by different methods for reasoning. For each trained model, we let it generate $I$ rules with highest qualities per query relation, and use them to train a predictor *w/o emb.* as in Eq. (5) for reasoning. For RNNLogic, the quality of each rule is measured by its prior probability from the rule generator, and we use beam search to infer top-$I$ rules. The results at different $I$ are in Fig. 2, where RNNLogic achieves much better results. Even with only 10 rules per relation, RNNLogic still achieves competitive results.

**Performance w.r.t. the Number of Training Triplets.** To better evaluate different methods under cases where training triplets are very limited, in this section we reduce the amount of training data on Kinship and UMLS to see how the performance varies. The results are presented in Fig. 4. We see that RNNLogic *w/o emb.* achieves the best results. Besides, the improvement over RotatE is more significant as we reduce training triplets, showing that RNNLogic is more robust to data sparsity.

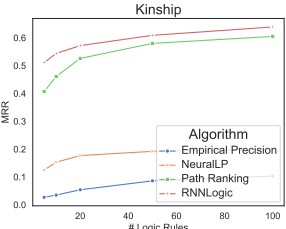 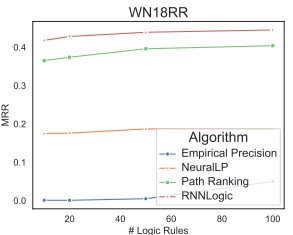 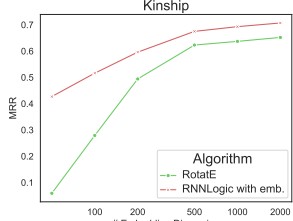

Figure 2: Performance w.r.t. # logic rules. RNNLogic achieves competitive results even with 10 rules per query relation.

Figure 3: Performance w.r.t. embedding dimension.

**Performance w.r.t. Embedding Dimension.** RNNLogic *with emb.* uses entity and relation embeddings to improve the reasoning predictor. Next, we study its performance with different embedding dimensions. The results are presented in Fig. 3, where we compare against RotatE (Sun et al., 2019). We see that RNNLogic significantly outperforms RotatE at every embedding dimension. The improvement is mainly from the use of logic rules, showing that our learned rules are indeed helpful.

**Comparison of Optimization Algorithms.** RNNLogic uses an EM algorithm to optimize the rule generator. In practice, the generator can also be optimized with REINFORCE (Williams, 1992) (see App. F for details). We empirically compare the two algorithms in the *w/o emb.* case. The results on Kinship and UMLS are presented in Tab. 4. We see EM consistently outperforms REINFORCE.

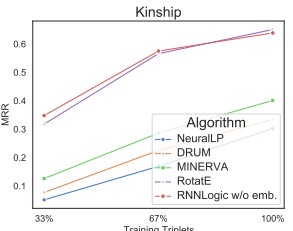
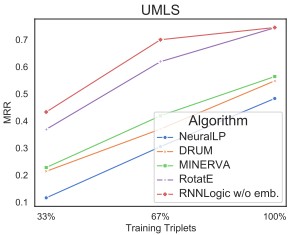

|  | Kinship MRR | UMLS MRR |
|---|---|---|
| REINFORCE | 0.312 | 0.504 |
| EM | 0.639 | 0.745 |

Table 4: Comparison between REINFORCE and EM.

Figure 4: Performance w.r.t. # training triplets. RNNLogic is more robust to data sparsity even without using embeddings.

**Case Study of Generated Logic Rules.** Finally, we show some logic rules generated by RNNLogic on the FB15k-237 dataset in Tab. 5. We can see that these logic rules are meaningful and diverse. The first rule is a subrelation rule. The third and fifth rules are two-hop compositional rules. The rest of logic rules have even more complicated forms. This case study shows that RNNLogic can indeed learn useful and diverse rules for reasoning. For more generated logic rules, please refer to App. E.

Table 5: Case study of the rules generated by the rule generator.

| |
|---|
| $\texttt{Appears\_in\_TV\_Show}(X,Y) \leftarrow \texttt{Actor\_of}(X,Y)$ |
| $\texttt{Appears\_in\_TV\_Show}(X,Y) \leftarrow \texttt{Creator\_of}(X,U) \wedge \texttt{Has\_Producer}(U,V) \wedge \texttt{Appears\_in\_TV\_Show}(V,Y)$ |
| $\texttt{ORG.\_in\_State}(X,Y) \leftarrow \texttt{ORG.\_in\_City}(X,U) \wedge \texttt{City\_Locates\_in\_State}(U,Y)$ |
| $\texttt{ORG.\_in\_State}(X,Y) \leftarrow \texttt{ORG.\_in\_City}(X,U) \wedge \texttt{Address\_of\_PERS.}(U,V) \wedge \texttt{Born\_in}(V,W) \wedge \texttt{Town\_in\_State}(W,Y)$ |
| $\texttt{Person\_Nationality}(X,Y) \leftarrow \texttt{Born\_in}(X,U) \wedge \texttt{Place\_in\_Country}(U,Y)$ |
| $\texttt{Person\_Nationality}(X,Y) \leftarrow \texttt{Student\_of\_Educational\_Institution}(X,U) \wedge \texttt{ORG.\_Endowment\_Currency}(U,V) \wedge$ $\texttt{Currency\_Used\_in\_Region}(V,W) \wedge \texttt{Region\_in\_Country}(W,Y)$ |

## 5 CONCLUSION

This paper studies learning logic rules for knowledge graph reasoning, and an approach called RNNLogic is proposed. RNNLogic treats a set of logic rules as a latent variable, and a rule generator as well as a reasoning predictor with logic rules are jointly learned. We develop an EM-based algorithm for optimization. Extensive expemriments prove the effectiveness of RNNLogic. In the future, we plan to study generating more complicated logic rules rather than only compositional rules. Besides, we plan to extend RNNLogic to other reasoning problems, such as question answering.

## ACKNOWLEDGMENTS

This project is supported by the Natural Sciences and Engineering Research Council (NSERC) Discovery Grant, the Canada CIFAR AI Chair Program, collaboration grants between Microsoft Research and Mila, Samsung Electronics Co., Ldt., Amazon Faculty Research Award, Tencent AI Lab Rhino-Bird Gift Fund and a NRC Collaborative R&D Project (AI4D-CORE-06). This project was also partially funded by IVADO Fundamental Research Project grant PRF-2019-3583139727.

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

# A  PROOFS

This section presents the proofs of some propositions used in the optimization algorithm of RNNLogic.

Recall that in the E-step of the optimization algorithm, we aim to sample from the posterior distribution over rule sets. However, directly sampling from the posterior distribution is intractable due to the intractable partition function. Therefore, we introduce Proposition 1, which gives an approximation distribution with more tractable form for the posterior distribution. With this approximation, sampling becomes much easier. In Section A.1, we present the proof of Proposition 1. In Section A.2, we show how to perform sampling based on the approximation of the posterior distribution.

## A.1  PROOF OF PROPOSITION 1

Next, we prove proposition 1, which is used to approximate the true posterior probability in the E-step of optimization. We first restate the proposition as follows:

**Proposition** *Consider a data instance $(\mathcal{G}, \boldsymbol{q}, \boldsymbol{a})$ with $\boldsymbol{q} = (h, r, ?)$ and $\boldsymbol{a} = t$. For a set of rules $\hat{\boldsymbol{z}}$ generated by the rule generator $p_\theta$, we can compute the following score $H$ for each rule $\in \hat{\boldsymbol{z}}$:*

$$H(rule) = \left\{ \text{score}_w(t|rule) - \frac{1}{|\mathcal{A}|} \sum_{e \in \mathcal{A}} \text{score}_w(e|rule) \right\} + \log \text{RNN}_\theta(rule|r),$$

*where $\mathcal{A}$ is the set of all candidate answers discovered by rules in $\hat{\boldsymbol{z}}$, $\text{score}_w(e|rule)$ is the score that each rule contributes to entity $e$ as defined by Eq. (4), $\text{RNN}_\theta(rule|r)$ is the prior probability of rule computed by the generator. Suppose $s = \max_{e \in \mathcal{A}} |\text{score}_w(e)| < 1$. Then for a subset of rules $\boldsymbol{z}_I \subset \hat{\boldsymbol{z}}$ with $|\boldsymbol{z}_I| = K$, the log-probability $\log p_{\theta,w}(\boldsymbol{z}_I | \mathcal{G}, \boldsymbol{q}, \boldsymbol{a})$ could be approximated as follows:*

$$\left| \log p_{\theta,w}(\boldsymbol{z}_I | \mathcal{G}, \boldsymbol{q}, \boldsymbol{a}) - \left( \sum_{rule \in \boldsymbol{z}_I} H(rule) + \gamma(\boldsymbol{z}_I) + \text{const} \right) \right| \le s^2 + O(s^4)$$

*where $\text{const}$ is a constant term that is independent from $\boldsymbol{z}_I$, $\gamma(\boldsymbol{z}_I) = \log(K! / \prod_{rule \in \hat{\boldsymbol{z}}} n_{rule}!)$, with $K$ being the given size of set $\boldsymbol{z}_I$ and $n_{rule}$ being the number of times each rule appears in $\boldsymbol{z}_I$.*

**Proof:** We first rewrite the posterior probability as follows:

$$\log p_{\theta,w}(\boldsymbol{z}_I | \mathcal{G}, \boldsymbol{q}, \boldsymbol{a}) = \log p_w(\boldsymbol{a} | \mathcal{G}, \boldsymbol{q}, \boldsymbol{z}_I) + \log p_\theta(\boldsymbol{z}_I | \boldsymbol{q}) + \text{const}$$

$$= \log \frac{\exp(\text{score}_w(t))}{\sum_{e \in \mathcal{A}} \exp(\text{score}_w(e))} + \log \text{Mu}(\boldsymbol{z}_I | K, \text{RNN}_\theta(\cdot|r)) + \text{const},$$

where const is a constant term which does not depend on the choice of $\boldsymbol{z}_I$, and $\text{RNN}_\theta(\cdot|r)$ defines a probability distribution over all the composition-based logic rules. The probability mass function of the above multinomial distribution $\text{Mu}(\boldsymbol{z}_I | K, \text{RNN}_\theta(\cdot|r))$ can be written as below:

$$\text{Mu}(\boldsymbol{z}_I | K, \boldsymbol{q}) = \frac{K!}{\prod_{rule \in \hat{\boldsymbol{z}}} n_{rule}!} \prod_{rule \in \hat{\boldsymbol{z}}} \text{RNN}_\theta(rule|r)^{n_{rule}},$$

where $K$ is the size of set $\boldsymbol{z}_I$ and $n_{rule}$ is the number of times a *rule* appears in $\boldsymbol{z}_I$.

Letting $\gamma(\boldsymbol{z}_I) = \log(K! / \prod_{rule \in \hat{\boldsymbol{z}}} n_{rule}!)$, then the posterior probability can then be rewritten as:

$$\log p_{\theta,w}(\boldsymbol{z}_I | \mathcal{G}, \boldsymbol{q}, \boldsymbol{a})$$

$$= \log \frac{\exp(\text{score}_w(t))}{\sum_{e \in \mathcal{A}} \exp(\text{score}_w(e))} + \log \frac{K!}{\prod_{rule \in \hat{\boldsymbol{z}}} n_{rule}!} + \log \prod_{rule \in \hat{\boldsymbol{z}}} \text{RNN}_\theta(rule|r)^{n_{rule}} + \text{const}$$

$$= \log \frac{\exp(\text{score}_w(t))}{\sum_{e \in \mathcal{A}} \exp(\text{score}_w(e))} + \gamma(\boldsymbol{z}_I) + \sum_{rule \in \boldsymbol{z}_I} \log \text{RNN}_\theta(rule|r) + \text{const}$$

$$= \text{score}_w(t) - \log \sum_{e \in \mathcal{A}} \exp(\text{score}_w(e)) + \gamma(\boldsymbol{z}_I) + \sum_{rule \in \boldsymbol{z}_I} \log \text{RNN}_\theta(rule|r) + \text{const}.$$

The above term $\log \sum_{e \in \mathcal{A}} \exp(\text{score}_w(e))$ makes the posterior distribution hard to deal with, and thus we approximate it using Lemma 1, which we prove at the end of this section.

**Lemma 1** *Let $e \in \mathcal{A}$ be a finite set of entities, let $|\texttt{score}_w(e)| \leq s < 1$, and let $\texttt{score}_w$ be a function from entities to real numbers. Then the following inequalities hold:*

$$0 \leq \log\left(\sum_{e \in \mathcal{A}} \exp(\texttt{score}_w(e))\right) - \left(\sum_{e \in \mathcal{A}} \frac{1}{|\mathcal{A}|}\texttt{score}_w(e) + \log(|\mathcal{A}|)\right) \leq s^2 + O(s^4).$$

Hence, using the lemma we can get the following upper bound of the posterior probability:

$$\log p_{\theta,w}(z_I|\mathcal{G}, q, a)$$
$$= \texttt{score}_w(t) - \log \sum_{e \in \mathcal{A}} \exp(\texttt{score}_w(e)) + \gamma(z_I) + \sum_{rule \in z_I} \log \text{RNN}_\theta(rule|r) + \text{const}$$
$$\leq \texttt{score}_w(t) - \sum_{e \in \mathcal{A}} \frac{1}{|\mathcal{A}|}\texttt{score}_w(e) + \gamma(z_I) + \sum_{rule \in z_I} \log \text{RNN}_\theta(rule|r) + \text{const}$$
$$= \sum_{rule \in z_I} H(rule) + \gamma(z_I) + \text{const},$$

and also the following lower bound of the posterior probability:

$$\log p_{\theta,w}(z_I|\mathcal{G}, q, a)$$
$$= \texttt{score}_w(t) - \log \sum_{e \in \mathcal{A}} \exp(\texttt{score}_w(e)) + \gamma(z_I) + \sum_{rule \in z_I} \log \text{RNN}_\theta(rule|r) + \text{const}$$
$$\geq \texttt{score}_w(t) - \sum_{e \in \mathcal{A}} \frac{1}{|\mathcal{A}|}\texttt{score}_w(e) + \gamma(z_I) + \sum_{rule \in z_I} \log \text{RNN}_\theta(rule|r) + \text{const} - s^2 - O(s^4)$$
$$= \sum_{rule \in z_I} H(rule) + \gamma(z_I) + \text{const} - s^2 - O(s^4),$$

where const is a constant term which does not depend on $z_I$.

By combining the lower and the upper bound, we get:

$$\left| \log p_{\theta,w}(z_I|\mathcal{G}, q, a) - \left(\sum_{rule \in z_I} H(rule) + \gamma(z_I) + \text{const}\right) \right| \leq s^2 + O(s^4)$$

Thus, it only remains to prove Lemma 1 to complete the proof. We use Theorem 1 from (Simic, 2008) as a starting point:

**Theorem 1** *Suppose that $\tilde{x} = \{x_i\}_{i=1}^n$ represents a finite sequence of real numbers belonging to a fixed closed interval $I = [a, b]$, $a < b$. If $f$ is a convex function on $I$, then we have that:*

$$\frac{1}{n}\sum_{i=1}^n f(x_i) - f\left(\frac{1}{n}\sum_{i=1}^n x_i\right) \leq f(a) + f(b) - 2f\left(\frac{a+b}{2}\right).$$

As $(-\log)$ is convex and $\exp(\texttt{score}_w(e)) \in [\exp(-s), \exp(s)]$, Theorem 1 gives us that:

$$-\frac{1}{|\mathcal{A}|}\sum_{e \in \mathcal{A}} \log\left(\exp(\texttt{score}_w(e))\right) + \log\left(\frac{1}{|\mathcal{A}|}\sum_{e \in \mathcal{A}} \exp(\texttt{score}_w(e))\right)$$
$$\leq -\log(\exp(-s)) - \log(\exp(s)) + 2\log\left(\frac{\exp(-s) + \exp(s)}{2}\right).$$

After some simplification, we get:

$$\log\left(\sum_{e\in\mathcal{A}}\exp(\texttt{score}_w(e))\right)$$

$$\leq \sum_{e\in\mathcal{A}}\frac{1}{|\mathcal{A}|}\texttt{score}_w(e) + \log(|\mathcal{A}|) + 2\log\left(\frac{\exp(-s)+\exp(s)}{2}\right) \tag{10}$$

$$= \sum_{e\in\mathcal{A}}\frac{1}{|\mathcal{A}|}\texttt{score}_w(e) + \log(|\mathcal{A}|) + 2s - 2\log 2 + 2\log(1+\exp(-2s))$$

$$\leq \sum_{e\in\mathcal{A}}\frac{1}{|\mathcal{A}|}\texttt{score}_w(e) + \log(|\mathcal{A}|) + s^2 + O(s^4),$$

where the last inequality is based on Taylor's series $\log(1+e^x) = \log 2 + \frac{1}{2}x + \frac{1}{8}x^2 + O(x^4)$ with $|x| < 1$. On the other hand, according to the well-known Jensen's inequality, we have:

$$\log\left(\frac{1}{|\mathcal{A}|}\sum_{e\in\mathcal{A}}\exp(\texttt{score}_w(e))\right) \geq \frac{1}{|\mathcal{A}|}\sum_{e\in\mathcal{A}}\log\left(\exp(\texttt{score}_w(e))\right),$$

which implies:

$$\log\left(\sum_{e\in\mathcal{A}}\exp(\texttt{score}_w(e))\right) \geq \sum_{e\in\mathcal{A}}\frac{1}{|\mathcal{A}|}\texttt{score}_w(e) + \log(|\mathcal{A}|). \tag{11}$$

By combining Eq. (10) and Eq. (11), we obtain:

$$0 \leq \log\left(\sum_{e\in\mathcal{A}}\exp(\texttt{score}_w(e))\right) - \left(\sum_{e\in\mathcal{A}}\frac{1}{|\mathcal{A}|}\texttt{score}_w(e) + \log(|\mathcal{A}|)\right) \leq s^2 + O(s^4).$$

This completes the proof.

$$\square.$$

## A.2 SAMPLING BASED ON THE APPROXIMATION OF THE TRUE POSTERIOR

Based on Proposition 1, the log-posterior probability $\log p_{\theta,w}(z_I|\mathcal{G}, q, a)$ could be approximated by $(\sum_{rule\in z_I} H(rule) + \gamma(z_I) + \text{const})$, with const being a term that does not depend on $z_I$. This implies that we could construct a distribution $q(z_I) \propto \exp(\sum_{rule\in z_I} H(rule) + \gamma(z_I))$ to approximate the true posterior, and draw samples from $q$ as approximation to the real samples from the posterior.

It turns out that the distribution $q(z_I)$ is a multinomial distribution. To see that, we rewrite $q(z_I)$ as:

$$q(z_I) = \frac{1}{Z}\exp\left(\sum_{rule\in z_I} H(rule) + \gamma(z_I)\right)$$

$$= \frac{1}{Z}\exp\left(\gamma(z_I)\right)\prod_{rule\in z_I}\exp\left(H(rule)\right)$$

$$= \frac{1}{Z}\frac{K!}{\prod_{rule\in\hat{z}} n_{rule}!}\prod_{rule\in\hat{z}}\exp\left(H(rule)\right)^{n_{rule}}$$

$$= \frac{1}{Z'}\frac{K!}{\prod_{rule\in\hat{z}} n_{rule}!}\prod_{rule\in\hat{z}} q_r(rule)^{n_{rule}}$$

$$= \frac{1}{Z'}\text{Mu}(z_I|K, q_r),$$

where $n_{rule}$ is the number of times a *rule* appears in the set $z_I$, $q_r$ is a distribution over all the generated logic rules $\hat{z}$ with $q_r(rule) = \exp(H(rule))/\sum_{rule'\in\hat{z}}\exp(H(rule'))$, $Z$ and $Z'$ are normalization terms. By summing over $z_I$ on both sides of the above equation, we obtain $Z' = 1$, and hence:

$$q(z_I) = \text{Mu}(z_I|K, q_r).$$

To sample from such a multinomial distribution, we could simply sample $K$ rules independently from the distribution $q_r$, and form a sample $\hat{z}_I$ with these $K$ rules.

In practice, we observe that the hard-assignment EM algorithm (Koller & Friedman, 2009) works better than the standard EM algorithm despite the reduced theoretical guarantees. In the hard-assignment EM algorithm, we need to draw a sample $\hat{z}_I$ with the maximum posterior probability. Based on the above approximation $q(z_I)$ of the true posterior distribution $p_{\theta,w}(z_I|\mathcal{G}, q, a)$, we could simply construct such a sample $\hat{z}_I$ with $K$ rules which have the maximum probability under the distribution $q_r$. By definition, we have $q_r(rule) \propto \exp(H(rule))$, and hence drawing $K$ rules with maximum probability under $q_r$ is equivalent to choosing $K$ rules with the maximum $H$ values.

## B  More Analysis of the EM Algorithm

In RNNLogic, we use an EM algorithm to optimize the rule generator. In this section, we show why this EM algorithm is able to maximize the objective function of the rule generator.

Recall that for a fixed reasoning predictor $p_w$, we aim to update $p_\theta$ to maximize the log-likelihood function $\log p_{w,\theta}(a|\mathcal{G}, q)$ for each data instance $(\mathcal{G}, q, a)$. Directly optimizing $\log p_{w,\theta}(a|\mathcal{G}, q)$ is difficult due to the latent logic rules, and therefore we consider the following evidence lower bound of the log-likelihood function:

$$\log p_{w,\theta}(a|\mathcal{G}, q) \geq \mathbb{E}_{q(z_I)}[\log p_w(a|\mathcal{G}, q, z_I) + \log p_\theta(z_I|q) - \log q(z_I)] = \mathcal{L}_{\text{ELBO}}(q, p_\theta), \quad (12)$$

where $q(z_I)$ is a variational distribution, and the equation holds when $q(z_I) = p_{\theta,w}(z_I|\mathcal{G}, q, a)$.

With this lower bound, we can optimize the log-likelihood function $\log p_{w,\theta}(a|\mathcal{G}, q)$ with an E-step and an M-step. In the E-step, we optimize $q(z_I)$ to maximize $\mathcal{L}_{\text{ELBO}}(q, p_\theta)$, which is equivalent to minimizing $\text{KL}(q(z_I)||p_{\theta,w}(z_I|\mathcal{G}, q, a))$. By doing so, we are able to tighten the lower bound. Then in the M-step, we further optimize $\theta$ to maximize $\mathcal{L}_{\text{ELBO}}(q, p_\theta)$. Next, we introduce the details.

**E-step.** In the E-step, our goal is to update $q$ to minimize $\text{KL}(q(z_I)||p_{\theta,w}(z_I|\mathcal{G}, q, a))$. However, there are a huge number of possible logic rules, and hence optimizing $q(z_I)$ on every possible rule set $z_I$ is intractable. To solve the problem, recall that we generate a set of logic rules $\hat{z}$ when optimizing the reasoning predictor, and here we add a constraint to $q$ based on $\hat{z}$. Specifically, we constrain the sample space of $q(z_I)$ to be all subsets of $\hat{z}$ with size being $K$, i.e., $z_I \subset \hat{z}$ and $|z_I| = K$. In other words, we require $\sum_{z_I \subset \hat{z}, |z_I|=K} q(z_I) = 1$. With such a constraint, we can further use proposition 1 to construct the variational distribution $q$ to approximate $p_{\theta,w}(z_I|\mathcal{G}, q, a)$, as what is described in the model section.

**M-step.** In the M-step, our goal is to update $p_\theta$ to maximize the lower bound $\mathcal{L}_{\text{ELBO}}(q, p_\theta)$. To do that, we notice that there is an expectation operation with respect to $q(z_I)$ in $\mathcal{L}_{\text{ELBO}}(q, p_\theta)$. By drawing a sample from $q(z_I)$, $\mathcal{L}_{\text{ELBO}}(q, p_\theta)$ can be estimated as follows:

$$\begin{aligned}
\mathcal{L}_{\text{ELBO}}(q, p_\theta) &= \mathbb{E}_{q(z_I)}[\log p_w(a|\mathcal{G}, q, z_I) + \log p_\theta(z_I|q) - \log q(z_I)] \\
&\simeq \log p_w(a|\mathcal{G}, q, \hat{z}_I) + \log p_\theta(\hat{z}_I|q) - \log q(\hat{z}_I),
\end{aligned} \quad (13)$$

where $\hat{z}_I \sim q(z_I)$ is a sample drawn from the variational distribution. By ignoring the terms which are irrelevant to $p_\theta$, we obtain the following objective function for $\theta$:

$$\log p_\theta(\hat{z}_I|q), \quad (14)$$

which is the same as the objective function described in the model section.

As a result, by performing the E-step and the M-step described in the model section, we are able to update $p_\theta$ to increase the lower bound $\mathcal{L}_{\text{ELBO}}(q, p_\theta)$, and thereby push up the log-likelihood function $\log p_{w,\theta}(a|\mathcal{G}, q)$. Therefore, we see that the EM algorithm can indeed maximize $\log p_{w,\theta}(a|\mathcal{G}, q)$ with respect to $p_\theta$.

## C  Details about Parameterization and Implementation

Section 3.2 of the paper introduces the high-level idea of the reasoning predictor with logic rules and the rule generator. Due to the limited space, some details of the models are not covered. In this section, we explain the details of the reasoning predictor and the rule generator.

## C.1 Reasoning Predictor with Logic Rules

We start with the reasoning predictor with logic rules. Recall that for each query, our reasoning predictor leverages a set of logic rules $\boldsymbol{z}$ to give each candidate answer a score, which is further used to predict the correct answer from all candidates.

Specifically, let $\mathcal{A}$ denote the set of all the candidate answers discovered by logic rules in set $\boldsymbol{z}$. For each candidate answer $e \in \mathcal{A}$, we define the following function $\texttt{score}_w$ to compute a score:

$$\texttt{score}_w(e) = \sum_{rule \in \boldsymbol{z}} \texttt{score}_w(e|rule) = \sum_{rule \in \boldsymbol{z}} \sum_{path \in \mathcal{P}(h,rule,e)} \psi_w(rule) \cdot \phi_w(path), \quad (15)$$

where $\mathcal{P}(h, rule, e)$ is the set of grounding paths which start at $h$ and end at $e$ following a *rule* (e.g., *Alice* $\xrightarrow{\texttt{friend}}$ *Bob* $\xrightarrow{\texttt{hobby}}$ *Sing*). $\psi_w(rule)$ and $\phi_w(path)$ are scalar weights of each *rule* and *path*.

For the scalar weight $\psi_w(rule)$ of a *rule*, we initialize $\psi_w(rule)$ as follows:

$$\psi_w(rule) = \mathbb{E}_{(\mathcal{G},\boldsymbol{q},\boldsymbol{a})\sim p_{\text{data}}} \left[ |\mathcal{P}(h, rule, t)| - \frac{1}{|\mathcal{A}|} \sum_{e \in \mathcal{A}} |\mathcal{P}(h, rule, e)| \right], \quad (16)$$

where $|\mathcal{P}(h, rule, t)|$ is the number of relational paths starting from head entity $h$, following the relations in *rule* and ending at tail entity $t$. The form is very similar to the definition of $H$ values for logic rules, and the value can effectively measure the contribution of a rule to the correct answers. We also try randomly initializing rule weights or initializing them as 0, which yield similar results.

For the scalar score $\phi_w(path)$ of a *path*, we either fix it to 1, or compute it by introducing entity and relation embeddings. In the second case, we introduce an embedding for each entity and relation in the complex space. Formally, the embedding of an entity $e$ is denoted as $\boldsymbol{x}_e$, and the embedding of a relation $\texttt{r}$ is denoted as $\boldsymbol{x}_\texttt{r}$. For a grounding path $path = e_0 \xrightarrow{\texttt{r}_1} e_1 \xrightarrow{\texttt{r}_2} e_2 \cdots e_{l-1} \xrightarrow{\texttt{r}_l} e_l$, we follow the idea in RotatE (Sun et al., 2019) and compute $\phi_w(path)$ in the following way:

$$\phi_w(path) = \sigma(\delta - d(\boldsymbol{x}_{e_0} \circ \boldsymbol{x}_{\texttt{r}_1} \circ \boldsymbol{x}_{\texttt{r}_2} \circ \cdots \circ \boldsymbol{x}_{\texttt{r}_l}, \boldsymbol{x}_{e_l})), \quad (17)$$

where $\sigma(x) = \frac{1}{1+e^{-x}}$ is the sigmoid function, $d$ is a distance function between two complex vectors, $\delta$ is a hyperparameter, and $\circ$ is the Hadmard product in complex spaces, which could be viewed as a rotation operator. Intuitively, for the embedding $\boldsymbol{x}_{e_0}$ of entity $e_0$, we rotate $\boldsymbol{x}_{e_0}$ by using the rotation operators defined by $\{\texttt{r}_k\}_{k=1}^l$, yielding $(\boldsymbol{x}_{e_0} \circ \boldsymbol{x}_{\texttt{r}_1} \circ \boldsymbol{x}_{\texttt{r}_2} \circ \cdots \circ \boldsymbol{x}_{\texttt{r}_l})$. Then we compute the distance between the new embedding and the embedding $\boldsymbol{x}_{e_l}$ of entity $e_l$, and further convert the distance to a value between 0 and 1 by using the sigmoid function and a hyperparameter $\delta$.

## C.2 Rule Generator

This paper focuses on compositional rules, which have the abbreviation form $\texttt{r} \leftarrow \texttt{r}_1 \wedge \cdots \wedge \texttt{r}_l$ and thus could be viewed a sequence of relations $[\texttt{r}, \texttt{r}_1, \texttt{r}_2 \cdots \texttt{r}_l, \texttt{r}_{\text{END}}]$, where $\texttt{r}$ is the query relation or the head of the rule, $\{\texttt{r}_i\}_{i=1}^l$ are the body of the rule, and $\texttt{r}_{\text{END}}$ is a special relation indicating the end of the relation sequence. We introduce a rule generator $\text{RNN}_\theta$ parameterized with an LSTM (Hochreiter & Schmidhuber, 1997) to model such sequences. Given the current relation sequence $[\texttt{r}, \texttt{r}_1, \texttt{r}_2 \cdots \texttt{r}_i]$, $\text{RNN}_\theta$ aims to generate the next relation $\texttt{r}_{i+1}$ and meanwhile output the probability of $\texttt{r}_{i+1}$. The detailed computational process towards the goal is summarized as follows:

- Initialize the hidden state of the $\text{RNN}_\theta$ as follows:

$$\boldsymbol{h}_0 = f(\boldsymbol{v}_\texttt{r}),$$

  where $\boldsymbol{v}_\texttt{r}$ is a parameter vector associated with the query relation or the head relation $\texttt{r}$, $f$ is a linear transformation.
- Sequentially compute the hidden state at different positions by using the LSTM gate:

$$\boldsymbol{h}_t = \text{LSTM}(\boldsymbol{h}_{t-1}, g([\boldsymbol{v}_\texttt{r}, \boldsymbol{v}_{\texttt{r}_t}]),$$

  where $\boldsymbol{v}_{\texttt{r}_t}$ is a parameter vector associated with the relation $\texttt{r}_t$, $[\boldsymbol{v}_\texttt{r}, \boldsymbol{v}_{\texttt{r}_t}]$ is the concatenation of $\boldsymbol{v}_\texttt{r}$ and $\boldsymbol{v}_{\texttt{r}_t}$, $g$ is a linear transformation.

- Generate $\mathbf{r}_{t+1}$ and its probability based on $\boldsymbol{h}_{t+1}$ and the following vector:

$$\text{softmax}(o(\boldsymbol{h}_{t+1})).$$

Suppose the set of relations is denoted as $\mathcal{R}$. We first transform $\boldsymbol{h}_{t+1}$ to a $|\mathcal{R}|$-dimensional vector by using a linear transformation $o$, and then apply softmax function to the $|\mathcal{R}|$-dimensional vector to get the probability of each relation. Finally, we generate $\mathbf{r}_{t+1}$ according to the probability vector.

## D    EXPERIMENTAL SETUP

### D.1    DATASET STATISTICS

The statistics of datasets are summarised in Table 6.

Table 6: Statistics of datasets.

| Dataset | #Entities | #Relations | #Train | #Validation | #Test |
|---|---|---|---|---|---|
| FB15K-237 | 14,541 | 237 | 272,115 | 17,535 | 20,466 |
| WN18RR | 40,943 | 11 | 86,835 | 3,034 | 3,134 |
| Kinship | 104 | 25 | 3,206 | 2,137 | 5,343 |
| UMLS | 135 | 46 | 1,959 | 1,306 | 3,264 |

### D.2    EXPERIMENTAL SETUP OF RNNLOGIC

Next, we explain the detailed experimental setup of RNNLogic. We try different configurations of hyperparameters on the validation set, and the optimal configuration is then used for testing. We report the optimal hyperparameter configuration as below.

**Data Preprocessing.** For each training triplet $(h, \mathbf{r}, t)$, we add an inverse triplet $(t, \mathbf{r}^{-1}, h)$ into the training set, yielding an augmented set of training triplets $\mathcal{T}$. To build a training instance from $p_{\text{data}}$, we first randomly sample a triplet $(h, \mathbf{r}, t)$ from $\mathcal{T}$, and then form an instance as $(\mathcal{G} = \mathcal{T} \setminus \{(h, \mathbf{r}, t)\}, \boldsymbol{q} = (h, \mathbf{r}, ?), \boldsymbol{a} = t)$. Basically, we use the sampled triplet $(h, \mathbf{r}, t)$ to construct the query and answer, and use the rest of triplets in $\mathcal{T}$ to form the background knowledge graph $\mathcal{G}$. During testing, the background knowledge graph $\mathcal{G}$ is constructed by using all the triplets in $\mathcal{T}$.

**Reasoning Predictor.** For the reasoning predictor in *with embedding* cases, the embedding dimension is set to 500 for FB15k-237, 200 for WN18RR, 2000 for Kinship and 1000 for UMLS. We pre-train these embeddings with RotatE (Sun et al., 2019). The hyperparameter $\delta$ for computing $\phi_w(path)$ in Equation (17) is set to 9 for FB15k-237, 6 for WN18RR, 0.25 for Kinship and 3 for UMLS. We use the Adam (Kingma & Ba, 2014) optimizer with an initial learning rate being $5 \times 10^{-5}$, and we decrease the learning rate in a cosine shape.

**Rule Generator.** For the rule generator, the maximum length of generated rules is set to 4 for FB15k-237, 5 for WN18RR, and 3 for Kinship and UMLS. These numbers are chosen according to model performace on the validation data. The size of input and hidden states in $\text{RNN}_\theta$ are set to 512 and 256. The learning rate is set to $1 \times 10^{-3}$ and monotonically decreased in a cosine shape. Beam search is used to generate rules with high probabilities, so that we focus on exploiting these logic rules which the rule generator is confident about. Besides, we pre-train the rule generator by using sampled relational paths on the background knowledge graph formed with training triplets, which prevents the rule generator from exploring meaningless logic rules in the beginning of training.

**EM Optimization.** During optimization, we sample 1000 rules from the rule generator for each data instance. In the E-step, for each data instance, we identify 300 rules as high-quality logic rules.

**Evaluation.** In testing, for each query $\boldsymbol{q} = (h, r, ?)$, we use the rule generator $p_\theta$ to generate 1000 logic rules, and let the reasoning predictor $p_w$ use the generated logic rules to predict the answer $\boldsymbol{a}$.

# E  CASE STUDY

We present more rules learned by RNNLogic on FB15k-237 dataset and UMLS dataset in Table 7 at the last page. In this table, $h \xrightarrow{\text{r}} t$ means triplet $(h, \text{r}, t)$ and $h \xleftarrow{\text{r}} t$ means triplet $(h, \text{r}^{-1}, t)$, or equivalently $(t, \text{r}, h)$.

# F  CONNECTION WITH REINFORCE

In terms of optimizing the rule generator, our EM algorithm has some connections with the RE-INFORCE algorithm (Williams, 1992). Formally, for a training instance $(\mathcal{G}, \boldsymbol{q}, \boldsymbol{a})$, REINFORCE computes the derivative with respect to the parameters $\theta$ of the rule generator as follows:

$$\mathbb{E}_{p_\theta(\boldsymbol{z}|\boldsymbol{q})}[R \cdot \nabla_\theta \log p_\theta(\boldsymbol{z}|\boldsymbol{q})] \simeq R \cdot \nabla_\theta \log p_\theta(\hat{\boldsymbol{z}}|\boldsymbol{q}), \tag{18}$$

where $\hat{\boldsymbol{z}}$ is a sample from the rule generator, i.e., $\hat{\boldsymbol{z}} \sim p_\theta(\boldsymbol{z}|\boldsymbol{q})$. $R$ is a reward from the reasoning predictor based on the prediction result on the instance. For example, we could treat the probability of the correct answer computed by the reasoning predictor as reward, i.e., $R = p_w(\boldsymbol{a}|\mathcal{G}, \boldsymbol{q}, \hat{\boldsymbol{z}})$.

In contrast, our EM optimization algorithm optimizes the rule generator with the objective function defined in Equation (2), yielding the derivative $\nabla_\theta \log p_\theta(\hat{\boldsymbol{z}}_I|\boldsymbol{q})$. Comparing the derivative to that in REINFORCE (Eq. (18)), we see that EM only maximizes the log-probability for rules selected in the E-step, while REINFORCE maximizes the log-probability for all generated rules weighted by the scalar reward $R$. Hence the two approaches coincide if $R$ is set to 1 for the selected rules from the approximate posterior and 0 otherwise. In general, finding an effective reward function to provide feedback is nontrivial, and we empirically compare these two optimization algorithms in experiments.

Table 7: Logic rules learned by RNNLogic.

| Relation | ← | Rule (*Explanation*) |
|---|---|---|
| $X \xrightarrow{\texttt{Appears\_in\_TV\_Show}} Y$ | ← | $X \xleftarrow{\texttt{Has\_Actor}} Y$ |
| | | (*Definition. An actor of a show appears in the show, obviously.*) |
| | ← | $X \xrightarrow{\texttt{Creator\_of}} U \xleftarrow{\texttt{Producer\_of}} V \xrightarrow{\texttt{Appears\_in\_TV\_Show}} Y$ |
| | | (*The creator X and the producer V of another show U are likely to appear in the same show Y.*) |
| | ← | $X \xleftarrow{\texttt{Actor\_of}} U \xleftarrow{\texttt{Award\_Nominated}} V \xleftarrow{\texttt{Winner\_of}} Y$ |
| | ← | $X \xrightarrow{\texttt{Writer\_of}} U \xleftarrow{\texttt{Creater\_of}} V \xrightarrow{\texttt{Actor\_of}} Y$ |
| | ← | $X \xrightarrow{\texttt{Student\_of}} U \xleftarrow{\texttt{Student\_of}} V \xrightarrow{\texttt{Appears\_in\_TV\_Show}} Y$ |
| | | (*Two students X and V in the same school U are likely to appear in the same show Y.*) |
| $X \xrightarrow{\texttt{ORG.\_in\_State}} Y$ | ← | $X \xrightarrow{\texttt{ORG.\_in\_City}} U \xrightarrow{\texttt{City\_in\_State}} Y$ |
| | | (*Use the city to indicate the state directly.*) |
| | ← | $X \xrightarrow{\texttt{ORG.\_in\_City}} U \xleftarrow{\texttt{Lives\_in}} V \xrightarrow{\texttt{Born\_in}} W \xrightarrow{\texttt{Town\_in\_State}} Y$ |
| | | (*Use the person living in the city to induct the state.*) |
| | ← | $X \xleftarrow{\texttt{Sub-ORG.\_of}} U \xrightarrow{\texttt{ORG.\_in\_State}} Y$ |
| | ← | $X \xrightarrow{\texttt{Sub-ORG.\_of}} U \xleftarrow{\texttt{Sub-ORG.\_of}} V \xrightarrow{\texttt{ORG.\_in\_State}} Y$ |
| | ← | $X \xrightarrow{\texttt{ORG.\_in\_City}} U \xleftarrow{\texttt{ORG.\_in\_City}} V \xrightarrow{\texttt{ORG.\_in\_State}} Y$ |
| | | (*Two organizations in the same city are in the same state.*) |
| $X \xrightarrow{\texttt{Person\_Nationality}} Y$ | ← | $X \xrightarrow{\texttt{Born\_in}} U \xrightarrow{\texttt{Place\_in\_Country}} Y$ |
| | | (*Definition.*) |
| | ← | $X \xrightarrow{\texttt{Spouse}} U \xrightarrow{\texttt{Person\_Nationality}} Y$ |
| | | (*By a fact that people are likely to marry a person of same nationality.*) |
| | ← | $X \xrightarrow{\texttt{Student\_of}} U \xrightarrow{\texttt{ORG.\_Endowment\_Currency}} V \xleftarrow{\texttt{Region\_Currency}} W \xrightarrow{\texttt{Region\_in\_Country}} Y$ |
| | | (*Use the currency to induct the nationality.*) |
| | ← | $X \xrightarrow{\texttt{Born\_in}} U \xleftarrow{\texttt{Born\_in}} V \xrightarrow{\texttt{Person\_Nationality}} Y$ |
| | ← | $X \xrightarrow{\texttt{Politician\_of}} U \xleftarrow{\texttt{Politician\_of}} V \xrightarrow{\texttt{Person\_Nationality}} Y$ |
| $X \xrightarrow{\texttt{Manifestation\_of}} Y$ | ← | $X \xleftarrow{\texttt{Treats}} U \xrightarrow{\texttt{Prevents}} V \xleftarrow{\texttt{Precedes}} Y$ |
| | ← | $X \xleftarrow{\texttt{Complicates}} U \xleftarrow{\texttt{Precedes}} Y$ |
| | ← | $X \xrightarrow{\texttt{Location\_of}} U \xrightarrow{\texttt{Is\_a}} V \xleftarrow{\texttt{Precedes}} Y$ |
| | ← | $X \xleftarrow{\texttt{Complicates}} U \xrightarrow{\texttt{Precedes}} V \xleftarrow{\texttt{Occurs\_in}} Y$ |
| | ← | $X \xrightarrow{\texttt{Location\_of}} U \xleftarrow{\texttt{Occurs\_in}} V \xleftarrow{\texttt{Occurs\_in}} Y$ |
| | ← | $X \xrightarrow{\texttt{Precedes}} U \xleftarrow{\texttt{Occurs\_in}} V \xleftarrow{\texttt{Degree\_of}} Y$ |
| $X \xleftarrow{\texttt{Affects}} Y$ | ← | $X \xrightarrow{\texttt{Result\_of}} U \xrightarrow{\texttt{Occurs\_in}} V \xrightarrow{\texttt{Precedes}} Y$ |
| | ← | $X \xleftarrow{\texttt{Precedes}} U \xrightarrow{\texttt{Produces}} V \xleftarrow{\texttt{Occurs\_in}} Y$ |
| | ← | $X \xleftarrow{\texttt{Prevents}} U \xrightarrow{\texttt{Disrupts}} V \xrightarrow{\texttt{Co-occurs\_with}} Y$ |
| | ← | $X \xleftarrow{\texttt{Result\_of}} U \xrightarrow{\texttt{Complicates}} V \xrightarrow{\texttt{Precedes}} Y$ |
| | ← | $X \xleftarrow{\texttt{Assesses\_Effect\_of}} U \xrightarrow{\texttt{Method\_of}} V \xrightarrow{\texttt{Complicates}} Y$ |
| | ← | $X \xrightarrow{\texttt{Process\_of}} U \xrightarrow{\texttt{Interacts\_with}} V \xrightarrow{\texttt{Causes}} Y$ |
| | ← | $X \xleftarrow{\texttt{Assesses\_Effect\_of}} U \xleftarrow{\texttt{Result\_of}} V \xrightarrow{\texttt{Precedes}} Y$ |

