# OpenReview forum: "RNNLogic: Learning Logic Rules for Reasoning on Knowledge Graphs"
_ICLR.cc/2021/Conference — ICLR 2021 Poster_

### Official Review · AnonReviewer2 · 2020-10-28

**Rating:** 7
**Confidence:** 4

**Review:**

In this paper, the author proposes RNNLogic for learning FOL rules from the knowledge graph. The proposed method assigns embeddings for each relation type and uses RNN module to generate chain-like rule candidates. Candidates are evaluated with a separate evaluation module that computes the scores. The rule scores are then taken to update the generator module using EM.

I think the idea of separating rule generation and evaluation is interesting, and using EM to jointly train the modules is also novel. However, I think the current paper writing is convoluted and prone to notations issues. One needs to constantly refer to appendix to understand the proposed method. For example:

- Why is Eq4 a valid answer score? I think it's related to the commonly-used graph traversal score, but the author should clarify this in the main text.

- Psi_w(rule) is not defined in the main text. And I found its definition in C1 to be problematic too: it's defined over score_w(t|rule) - but isn't that the score_w is defined with psi_w(rule) and psi_w(path) already in Eq4?

- I suggest the author to elaborate more why RotatE can be used to generate psi_w(path), as it's not very intuitive to me why this would help

Some claims are not well addressed:

- In section 2, the author criticizes the current differentiable ILP methods to lack ways "to determine the importance of rules with the learned weights due to the high dimensionality.". However, many methods such as diff-ILP and NeuralLP can indeed learn the rule weights.

- Backward-chaining methods such as NeuralLP are indeed efficient for problems with high dimensionality as well.

- There seems to be no justification or experiments to demonstrate how the proposed method can do better than the previous ones on these properties.

Model details:

- The author claims the psi_w(path) can be fixed to 1, in this case, what's been learned w.r.t parameter w? Is Psi_w(rule) a learnable module as well?

Model efficiency:

- Evaluating Eq 4 and Eq 5 requires to sum over all possible paths and rules. This process can be efficient for methods utilizing matrix multiplication such as NeuralLP. But it's unclear to me if a similar method is used in RNNLogic. If not, how well does the model scale to long paths and KBs with large grounding space?

- Also, the EM algorithm requires to hard sample top-K rules (K=1000) for each data instance at each iteration. I'm concerned about the efficiency of this method. It would be helpful if the author can provide the runtime comparison in the experiment as well.


Experiment:

- Many baseline scores on the FB15K and WN18 are cited from the original paper. But why do NeuralLP and DRUM get rerun in these twos benchmark?

- In appendix D, the author mentions that the entity embeddings are in fact *pre-trained* using RotatE. I find this setting to be unjustified and can lead to unfair comparisons against both ILP and non-ILP methods.

- In the w/o emb. mode, I assume no embeddings are learned in the reasoning module, so the score should be comparable to the vanilla score used in NeuralLP. Given that the RNNLogic also generates only chain-like rules using RNN, it seems that the proposed method should have searched similar rules and got similar rule scores as the NeuralLP. Can author provide intuitions or insights why RNNLogic significantly outperforms the NeuralLP even in w/o emb. mode?

Overall, I think this work indeed has some novelties but it is currently prone to unjustified claims and confusing writing. The experiment is incomplete and the usage of pre-trained embedding for SOTA model is unjustified. With that being said, I would recommend weak rejection at this point, but I'm happy to raise the scores if these concerns are addressed.

---

> ### Author Response · Authors · 2020-11-17
> **Response to the AnonReviewer2 cont.**
>
> --------------------------------------------------------------
> ##### **About the experiment.**
>
> **(1) About why NeuralLP and DRUM are rerun in FB15k-237 and WN18RR?**
>
> Thanks for pointing this out.  The reason is that NeuralLP and DRUM use a slightly different evaluation setup in experiment. To make the comparison more fair, we rerun these methods to make sure that the same evaluation setup is used for all the compared methods. For the detailed evaluation setup used in the experiment, please refer to the section of Evaluation Metrics in the original paper.
>
> **(2) About using pre-trained entity/relation embeddings.**
>
> For pre-training knowledge graphs, it is a common practice in the literature of knowledge graph reasoning. For example, some methods based on reinforcement learning use pre-trained knowledge graph embedding for reward shaping [1]. Some neural theorem prover methods and graph neural network methods also use pre-trained knowledge graph embedding for model initialization [2,3].
>
> The reason why we use pre-trained embedding in RNNLogic is that we aim to show RNNLogic can improve over existing knowledge graph embedding models by using logic rules. In fact, comparing RNNLogic with embedding and other embedding-based methods, we can see RNNLogic achieves comparable or even better results (table 1 and 2). In particular, RNNLogic with embedding significantly outperforms RotatE on Kinship and UMLS, which proves that RNNLogic can indeed learn high-quality logic rules to benefit SOTA embedding methods.
>
> To fairly compare RNNLogic with other non-embedding methods, especially those ILP methods, we can use the variant of RNNLogic without using embedding for comparison (i.e., RNNLogic w/o emb.). For this variant, we can see that it almost outperforms all the non-embedding methods, and even achieves comparable results to SOTA embedding-based methods, which proves the effectiveness of RNNLogic even without using embedding.
>
> [1] Lin, Xi Victoria, Richard Socher, and Caiming Xiong. "Multi-hop knowledge graph reasoning with reward shaping." arXiv preprint arXiv:1808.10568 (2018).
> [2] Rocktäschel, Tim, and Sebastian Riedel. "End-to-end differentiable proving." Advances in Neural Information Processing Systems. 2017.
> [3] Nathani, Deepak, et al. "Learning attention-based embeddings for relation prediction in knowledge graphs." arXiv preprint arXiv:1906.01195 (2019).
>
> **(3) About the comparison of RNNLogic w/o emb. and NeuralLP.**
>
> This is a good question! As you pointed out, the search space of NeuralLP and RNNLogic without embedding is the same if we use the same maximum rule length. For NeuralLP, it tries to learn rule weights jointly for all the logic rules in the search space through an attention mechanism, and the optimization can be hard due to the large number of logic rules. In contrast to NeuralLP, RNNLogic introduces a rule generator to learn a prior distribution of useful logic rules. In this way, the rule generator helps avoid considering those less useful logic rules, which reduces the search space and makes rule weight optimization much easier.
>
> To demonstrate the effect of the rule generator, we compare RNNLogic w/o emb., NeuralLP and a variant of RNNLogic. The variant only has a reasoning predictor, which learns rule weights for all the chain-like rules up to length 3, and uses these rules for reasoning. Therefore, the variant works in a similar way to NeuralLP. In contrast, RNNLogic uses a rule generator to parameterize a rule prior. Rather than considering all chain-like rules, the reasoning predictor in RNNLogic only learns rule weights for those rules generated by the rule generator, and uses such rules for reasoning. The results are presented as below:
>
> |                              | WN18RR MRR    | Kinship MRR  |
> | ---------------------------- |:-------------:|:------------:|
> | NeuralLP                     | 0.381         | 0.382        |
> | RNNLogic Variant (w/o emb.)  | 0.418         | 0.480        |
> | RNNLogic (w/o emb.)          | 0.455         | 0.639        |
>
> We can see that the variant of RNNLogic has similar results to NeuralLP, as they both use all chain-like rules up to length 3 for reasoning. For RNNLogic, it significantly outperforms the other methods. The reason is that RNNLogic only learns weights for those generated high-quality logic rules and uses them for reasoning, which makes the optimization process much easier and avoids the side effect of noisy logic rules during reasoning.
>
> --------------------------------------------------------------

---

> ### Author Response · Authors · 2020-11-17
> **Response to the AnonReviewer2 cont.**
>
> --------------------------------------------------------------
>
> ##### **About model details.**
>
> **(1) In this case, what's been learned w.r.t parameter w? Is $\psi_w(rule)$ a learnable module as well?**
>
> You are right! $\psi_w(rule)$ is the scalar weight of each logic rule, and it is a learnable parameter. The learnable modules in the reasoning predictor of RNNLogic are (i) the embedding of entities and relations; (ii) the scalar weight of each logic rule. We have clarified that in the updated draft (see the dark blue paragraph in page 4).
>
> --------------------------------------------------------------
>
> ##### **About model efficiency.**
>
> **(1) About whether matrix multiplication is used?**
>
> In our implementation, we tried two different ways to compute the summation over all possible paths and rules. One way is to use Breadth-First Search (BFS) to find all possible paths and then compute the summation. The other way is to use sparse matrix multiplication for computation, which is similar to the implementation of NeuralLP. On the four standard datasets, the two ways result in similar running time. The possible reason is that each time the rule generator only generates a small number of logic rules (e.g., 2000), and the maximum rule length is quite small on all the datasets (e.g., 4). Therefore, the total number of grounding paths is relatively small, and thus using matrix multiplication will not significantly improve efficiency.
>
> **(2) About hard sampling top-K rules in EM.**
>
> Our approach is not very sensitive to the choice of K, and we are able to achieve similar results when K is set to 300 or even 10. Also, no matter how we set K, we need to compute the H score (equation 7 of the paper) for every generated rule at each iteration, and thus the computational cost is almost the same for different K.
>
> **(3) About runtime comparison.**
>
> We conduct additional experiments to compare the runtime of NeuralLP and RNNLogic without using embedding on FB15k-237, WN18RR and Kinship. On all the datasets, we set the maximum rule length to 3. The prediction result (MRR) and running time (in minutes) are presented as below:
>
> | FB15k-237     | MRR           | Time  |
> | ------------- |:-------------:|:-----:|
> | NeuralLP      | 0.237         | 1358  |
> | RNNLogic      | 0.253         |  411  |
>
> | WN18RR        | MRR           | Time  |
> | ------------- |:-------------:|:-----:|
> | NeuralLP      | 0.381         |   67  |
> | RNNLogic      | 0.455         |   38  |
>
> | Kinship       | MRR           | Time  |
> | ------------- |:-------------:|:-----:|
> | NeuralLP      | 0.382         |    2  |
> | RNNLogic      | 0.639         |    9  |
>
> We can see that RNNLogic has similar running time to NeuralLP.
>
> --------------------------------------------------------------

---

> ### Author Response · Authors · 2020-11-17
> **Response to the AnonReviewer2**
>
> Thank you for the insightful comments, which are helpful for us to further improve the quality of the paper! Please find the answers to your questions as below:
>
> --------------------------------------------------------------
> ##### **About some issues in paper writing.**
> Due to the limited pages and spaces, we put some model details in the appendix and we are sorry about the confusion. We have fixed some presentation issues (see the blue texts in the updated draft), and we will keep refining the writing.
>
> **(1) About why is Eq4 a valid answer score?**
>
> Equation 4 is inspired by stochastic logic programming, which can be indeed understood as a kind of graph traversal score. Basically, for each logic rule, we can start from the head entity in a query, and traverse the knowledge graph according to the relations in the body of the logic rule, resulting in different ending entities as candidate answers. Then the score of each candidate answer is computed with both the weight of the rule and the specific relational path in the knowledge graph.
>
> **(2) About psi_w(rule).**
>
> Psi_w(rule) is the scalar weight of a rule, which is a learnable parameter. In practice, psi_w(rule) can be randomly initialized, zero initialized or initialized by using equation 16 in the appendix, and all these options yield similar results. The equation 16 in the original draft is problematic, and we have fixed the equation in the updated draft. Also, the definition of psi_w(rule) has been added in the main text (see the dark blue paragraph in page 4).
>
> **(3) About why RotatE can be used to generate psi_w(path)?**
>
> This is a very good question! Given a query, a logic rule can find different grounding paths, yielding different candidate answers. For example, for a query (Alice, hobby, ?), a logic rule hobby <= friend \and hobby can have two grounding paths, i.e., [Alice -friend- Bob -hobby- Sing] and [Alice -friend- Charlie -hobby- Ski], which give two candidate answers Sing and Ski.
>
> If we fix psi_w(path) to 1, then both grounding paths will be equally weighted, and the two candidate answers (Sing and Ski) will receive equal scores, meaning that we could not well distinguish between different candidate answers. By using RotatE, we can use the entity/relation embeddings to compute a path-specific score psi_w(path), which measures the soundness or consistency of each grounding path. In this way, we can better distinguish between different grounding paths and predict the answer more effectively.
>
> To be more specific, each relation in RotatE is modeled as a rotation operator in the entity embedding space. For a grounding path (e.g., [Alice -friend- Bob -hobby- Sing]), if we rotate the starting entity (e.g., Alice) according to the relations in the path (e.g., friend, hobby), the rotated embedding should be close to the ending entity (e.g., Sing). A smaller distance between the rotated embedding and the ending entity indicates that the grounding path has better soundness. Therefore, in our implementation, we use the distance between the rotated head entity embedding and the tail entity to compute psi_w(path). Similarly, one can also use the TransE model to parameterize psi_w(path).
>
> --------------------------------------------------------------
>
> ##### **About that some claims are not well addressed.**
>
> **(1) About many methods such as diff-ILP and NeuralLP can indeed learn the rule weights, and backward-chaining methods such as NeuralLP are indeed efficient for problems with high dimensionality as well.**
>
> You are right. Many ILP methods such as diff-ILP and NeuralLP are indeed able to learn the rule weights and are efficient for problems with high dimensionality as well. We have revised the claims over previous ILP methods in the updated version (see the dark blue text in the first paragraph of related work).
>
> As suggested by Reviewer 3, the main challenge of previous ILP methods is that they try to simultaneously generate the rules and learn their weights, which is difficult in terms of optimization. The main innovation of our method is to separate the process of rule generation and rule weight learning, which can be mutually enhanced with an EM algorithm. More specifically, the rule generation module generates high-quality rules, which significantly reduces the search pace, and hence allows the rule weight learning module to only focus on learning the weights of high-quality rules, yielding better results. Meanwhile, the rule weight learning module can in turn help identify some useful logic rules to improve the rule generator.
>
> In the experiment, some results can support our claim:
> a) In table 1 and 2, RNNLogic without embedding outperforms NeuralLP and DRUM.
> b) In figure 2, the top-ranked logic rules learned by RNNLogic yields better predictive results than NeuralLP, showing that RNNLogic can indeed learn better logic rules.
>
> --------------------------------------------------------------

---

### Official Review · AnonReviewer4 · 2020-10-30
**Marginal Gain over existing work**

**Rating:** 6
**Confidence:** 2

**Review:**

This paper focuses on learning logic rules via EM-based algorithm. The idea in the paper is that E-step would try to generate rules while M-step would update the parameters. The empirical comparisons are interesting and show that the paper improves on prior work.

There are several aspects that make it very hard for me to understand the paper's contributions and evaluation. I am listing below and hoping that other reviewers or authors would be able to clarify:

1. The paper suggests that the performance is owing to reduction of search space due to E-M step. It is hard to understand what MLN tool is being employed here and why MLN-based technique would return a suboptimal answer (the combinatorial solvers may return suboptimal answer due to timeout but that should be clarified in this case). It may perhaps be the case that sampling rules from a good distribution allows us to search only over a small space but that needs to contrasted with weakness of MLN-based methods.

2. The paper uses mean ranking for measuring. I am failing to understanding how is the ranking computed for rules and why such a metric is a good approach.  Can authors expand on what exactly is being done: i don't understand "for each query, we compute probability for each entity" means?

---

> ### Author Response · Authors · 2020-11-17
> **Response to the AnonReviewer4**
>
> Thank you for the insightful comments!
>
> --------------------------------------------------------------
> (1) For MLN, we implement it by ourselves. The implementation takes a fixed set of chain-like candidate logic rules as input and learns their rule weights for reasoning. We didn’t try those algorithms for MLN structure learning (i.e., learning the optimal set of logic rules for MLNs), where combinatorial solvers are involved, as these algorithms are inefficient.
>
> As a remedy, in the experiment, we compare against another more efficient structure learning method, boosted RDN (Relational Dependency Networks), and the tool we use is RDN-Boost (http://pages.cs.wisc.edu/~tushar/rdnboost/index.html). RDNs are similar to MLNs, as they both combine graphical models and first-order logic. The implementation uses functional gradient boosting, which allows to simultaneously learn both structure and parameters of RDNs efficiently. However, as the search space of all the possible RDN structures (i.e., all sets of logic rules) is huge, this more efficient combinatorial solver still cannot achieve good results in a reasonable amount of time.
>
> Compared with these methods, our approach introduces a rule generator to parameterize the distribution of useful logic rules. Such a rule generator serves as a prior, which filters out many less useful logic rules and thus reduces the search space, as you pointed out. We will further clarify that in the updated draft.
>
> --------------------------------------------------------------
> (2) For evaluation, we don’t compute the rank of different logic rules. Instead, we evaluate different methods by using a link prediction task, where the rank of entities is computed. To be more specific, given a query (h, r, ?), the task aims to infer the correct entity t, so that entity h and entity t have the relation r. To do that, we compute a score or probability for each entity in the entity vocabulary, and then sort all the entities according to the scores or probabilities. Afterwards, the rank of the correct answer is computed, based on which we further compute the mean rank over all the queries. Such an evaluation setup was proposed in [1], and is now widely used in knowledge graph reasoning.
>
> [1] Bordes, Antoine, et al. "Translating embeddings for modeling multi-relational data." Advances in neural information processing systems. 2013.
>
> --------------------------------------------------------------

---

> > ### Comment · AnonReviewer4 · 2020-11-23
> > **Thanks for the response**
> >
> > Apologies for the late response. Thank you for your response; I have increased my score as the answers convince something interesting is going on but need to be understood further.

---

> > > ### Author Response · Authors · 2020-11-24
> > > **Thanks for the comments**
> > >
> > > Thank you for the comments! We will try to further improve the quality of the paper.

---

### Official Review · AnonReviewer1 · 2020-10-31

**Rating:** 8
**Confidence:** 1

**Review:**

In this work, the authors illustrate an approach for learning logical rules starting from knowledge graphs. Learning logic rules is more interesting than simply performing link prediction because rules are human-readable and hence provide explainability.
The approach seems interesting and the tackled problem could interest a wide audience. It does not seem extremely novel, but it seems valid to me.
The paper is well-written and self-contained. Moreover, the experimental results show that the proposed approach has competitive performance comparing to other systems (even compared with systems that do not learn rules but perform only link prediction).

For all these reasons I think the paper should be accepted for publication.

---

> ### Author Response · Authors · 2020-11-17
> **Response to the AnonReviewer1**
>
> Thank you for the comments! We will try to further improve the quality of the paper.

---

### Official Review · AnonReviewer3 · 2020-11-01
**Review of "RNNLogic: Learning Logic Rules for Reasoning on Knowledge Graphs"**

**Rating:** 6
**Confidence:** 4

**Review:**

There is a lot of recent work on link-prediction in knowledge graphs. One approach is based on embedding entities and relations in a knowledge graph into vector spaces, and the other is based on finding rules that imply relations, and then using these rules to find new links or facts. This paper takes the latter approach. Within the area of rule-based methods, a number of recent papers have used neural network methods to simultaneously generate rules and to find rule-weights or other related parameters (indicating how important individual rules are). Simultaneously solving for rules and rule-weights is a difficult task. In this paper, the authors propose a method where they separate the rule generation process from the weight/parameter calculation process. More importantly, they add a feedback loop from the weight calculation routine ("reasoning predictor") to the rule generation routine ("rule generator"), which in my opinion is novel, even though there have been a few recent attempts (Xiong et. al. 2017) to use reinforcement learning to search for rules. The rule generator in this paper uses a recurrent neural network (RNN) and the parameters of this RNN are modified by the reasoning predictor. In other words, the iterative process has the feature that new rule generation is influenced by the calculated weights of previously generated rules. The authors perform numerical experiments on 4 standard knowledge graphs to demonstrate the performance of their method

Strong points

1. Clean probabilistic framework to jointly solve for rule generation and rule weight calculation via an iterative process where rule weight calculation (of existing rules) influences subsequent rule generation.

2. Good results

3. The authors take into account recent concerns on the quality of knowledge graph link prediction methods voiced in Sun et. al. (2020). Many papers use a ranking approach that yields excessively good scores to low-quality results (if many entities including the correct one receive equal scores as potential solutions to the query (h,r,?) during testing , where h is an entity and r is a relation, then the correct entity is given a high score), and the authors fix one of the issues in such ranking approaches.

Weak points

1. In many rule based methods (such as Neural LP), the goal is to find a set of (weighted) rules that imply a single relation.
In this paper, it seems that the authors generate a set of weighted rules for each "query" of the form (h,r, ?) derived from a fact/triplet of the form (h,r,t) where h,t are entities and r is a relation. My issues with this are:

 a) computation time, given that the number of relations maybe in the few hundreds (e.g. FB15k-237), whereas the number of facts/triplets is significantly larger. The authors do not say anything about computation time (this seems standard in the area).

 b) In the test-phase, it is not clear how one deals with entities which were not present in the training set? If one had a rule for a relation, then one would still be  able to use the rule.

3. Though the authors fix some issues with over-optimistic ranking approaches during testing, they do not clarify what happens when many entities receive equal non-zero scores. In other words, they do not explain what rank they give to the correct entity for the query (h,r,?) if it receives an equal probability score to a number of other entities.

4. The authors focus on the theoretical aspects of the paper. However, replicating the experiments in this paper seems to be difficult. The high-level ideas are easy to follow, but the precise way the algorithm is put together is hard to follow.

Recommendation

I recommend accepting this paper.
My acceptance recommendation is based on the "Strong Points" 1 and 3. The paper gives a well-founded approach to improving rule-generation based on previously calculated rule weights, and uses an improved scoring method, so that high scores are more meaningful than in some prior papers.

Questions

1. Is my understanding correct that you build rules for each query q = (h,r,?) and not for each relation? This seems to be implied in Figure 2. Please explain how you will deal with head/tail entities in the test set that are missing from the training set?

2. You use "reverse arcs" in training set, but you do not make clear if you use inverse relations in the rules you generate. Do you?

3. In the Sun et. al. paper, they talk about doing a "fair comparison" when an algorithm gives a certain probability/score to the correct answer (and to many incorrect answers) of a query. You only use their suggested fix for the case correct answers are given zero probabilities. What happens when multiplies entities are given identical nonzero probabilities?

4. Can you clarify what RNNLogic with embedding is? I am guessing it is related to assigning path scores via embedding? Please make this explicit.

5. Will your method be computationally more expensive than methods which generate rules for individual relations?

6. In trying to answer a query (h,r,?), it is clear you may use rules that include the relation 'r'. What is to stop you from using rules of the from 'r(X,Y) and s(Y,Z) and s^-1(Z,Y)'  and other similar rules which implicitly create trivial relational paths?

7. There are a number of papers in the area of rule-learning for simple binary classification tasks where rule weight calculation influences subsequent rule generation. See papers on boosting classifiers, especially boosting rule-based classifiers e.g. Eckstein and Goldberg (ICML 2010). You should probably give a reference to such work.

8. In the definition of the probability value in equation (5), you are implicitly assuming that if a rule creates multiple paths from a head entity h to multiple entities other than e (say b, and c), than the rule is not good for e. But what if (h,r,b) and (h,r,c) are facts in the training set in addition to (h,r,e)? Should you be penalizing this rule (or set of rules z)?

9. It seems to me that you are implicitly assuming that if a fact (h,r,e) is missing from the training set, then it is not true; in other words, a closed-world assumption. Is this correct? If so, it would help to mention it explicitly.

Though I like the paper, and I believe it has good results, I am now seriously concerned about the quality of the numbers in Tables 1 and 2. The origin of many of the results in these tables is in doubt. The authors say that the results for TransE, RotatE, ConvE, ComplEx , DistMult, and Minerva are taken from the corresponding papers. But when I look at these papers, I see different (or no) numbers.

The values for Minerva in the current paper match for FB15k-237, but do not match for WN18RR. The values in the Minerva paper for UMLS or Kinship are way better than in the current paper. Now Minerva uses a different evaluation protocol, but I seriously doubt the Minerva numbers for UMLS and Kinship.

RotatE has no numbers for UMLS or Kinship.

ComplEx has none of the numbers reported here.

Distmult has none of the numbers reported here.

The numbers in the current paper for TransE, DistMult, RotatE, ConvE and ComplEx seem to be taken from the RotatE paper. But RotatE has no numbers for UMLS or Kinship.

So overall, the provenance of the numbers in Table 1 and 2 is in serious doubt. I realize that the authors do not have a chance to respond and modify the paper. I hope the PC members can weigh in on what can be done at this stage. Even though I like this paper, my score will go down based on the poor quality of Tables 1 and 2.

---

> ### Author Response · Authors · 2020-11-17
> **Response to the AnonReviewer3**
>
> Thank you for the valuable suggestions!
>
> For your questions:
>
> (1) This is a good point! Our rule generator is formalized as $p_\theta(z|q)$ with $q = (h, r, ?)$. However, as you pointed out, generating logic rules conditioned on both $h$ and $r$ can be inefficient and unable to deal with entities not in the training set. Therefore, in our implementation, the rule generator generates logic rules only based on the query relation $r$, without considering the query entity $h$ (see the light blue paragraph in page 4). In other words, we assume that the generated logic rules $z$ and the query entity $h$ are independent when conditioned on the query relation $r$. For figure 2, more precisely we generate a number of logic rules for each relation $r$, and then apply these rules to all the queries with the form $q = (h, r, ?)$. We are sorry about the confusion and we have clarified that in the updated paper (see the light blue text in page 8).
>
> (2) We also use inverse relations in the generated rules. You may refer to section E and table 7 in the appendix for some concrete examples.
>
> (3) Thanks for pointing it out! Indeed, the chances that two entities receive equal non-zero probabilities are very small. For such cases, we also use the expected rank for evaluation. To be more specific, if there are $M$ entities receiving higher probabilities than the correct answer, and $N$ entities receiving the same probability as the correct answer, then the rank is computed as $(M + N / 2)$.
>
> (4) You are right. For RNNLogic with embedding, it computes path scores by using entity and relation embeddings. We have further clarified that in the updated draft. See the light blue sentence in the section of compared algorithms.
>
> (5) We are sorry for the confusion. As explained in question (1), in our implementation, given a query $q = (h, r, ?)$, we generate logic rules by only considering the query relation $r$ and ignoring the query entity $h$. Therefore, the method has similar computational cost to other methods which generate rules for individual relations.
>
> (6) Thanks for pointing it out. For now we don’t deal with the case in our generator, but avoiding such trivial relational paths should further reduce the search space. We will try this idea in the future by changing the implementation of rule grounding.
>
> (7) Thank you for pointing out the related work! We have cited and discussed the work in the updated draft. See the light blue paragraph in the related work section.
>
> (8) In practice, there are indeed cases where a query $q = (h, r, ?)$ can have multiple answers (e.g., $e$, $b$, $c$) according to the training set. For such cases, our implementation treats all of $e$, $b$, $c$ as correct answers, and maximizes $(\log p(e) + \log p(b) + \log p(c))$. Therefore, it will not penalize the rules you mentioned.
>
> (9) You are correct! We are using a closed-world assumption here. We have mentioned it explicitly in the updated draft (see the light blue text in page 7).

---

> > ### Comment · AnonReviewer3 · 2020-11-23
> > **Question regarding rank for entities receiving nonzero probabilities**
> >
> > This is in response to your answer (3):
> >
> > >> (3) Thanks for pointing it out! Indeed, the chances that two entities receive equal non-zero probabilities are very small. ...
> >
> > Could you point out the location in the code where you compute the rank of entities with non-zero probabilities as (M+N/2)?
> > I looked through the code and it is not clear to me. In particular, if you could point out the python file name and line number, along with the subdirectory it is in, I can check for myself.

---

> > > ### Author Response · Authors · 2020-11-24
> > > **Response to the question regarding rank for entities receiving nonzero probabilities**
> > >
> > > Thank you for the question!
> > >
> > > The code can be found in code/model.py at line 495-502, where rankl is the number of entities whose probabilities are greater than the correct answer, and rankr is the number of entities whose probabilities are greater or equal to the correct answer. Based on rankl and rankr, we use several loop-up tables (e.g., self.pre_mr) to compute the expectation of different metrics, and the look-up tables are defined at line 40-44.
> > >
> > > Besides, the description of the evaluation setup in the original paper has some ambiguity due to the limited space. We have further clarified that and discussed the case of "multiple entities are given identical nonzero probabilities" in the updated version of the paper (see the light blue text in the section of Evaluation Metrics).

---

> ### Author Response · Authors · 2021-01-17
> **Response to the AnonReviewer3 cont.**
>
> Thanks for your helpful comments regarding Tables 1 and 2!
>
> For Table 1, we made a mistake when copying the results of Minerva on WN18RR, so the numbers were not consistent with those reported in the Minerva paper. This problem has been fixed in the revised draft. We are sorry about our carelessness, and we are grateful to you for pointing it out.
>
> For the Kinship and UMLS datasets, we didn't find a common train/validation/test data split used by all existing works, so we followed several existing papers (e.g., NeuralLP, NTP) to create a data split by ourselves. When creating the data splits, we noticed that our approach achieved much better results than many baseline methods when the number of training facts is very small. We think this observation is helpful to better understand our approach, so we split the dataset into train/validation/test sets with a ratio 3:2:5, as explained in the Section 4.1. For this new split, we ran different baseline methods by ourselves and evaluated them with the same protocol, so those numbers couldn't be found in existing papers. We are sorry for the confusion and we will further clarify that in the caption of Table 2.
>
> Again, thanks for your helpful comments!

---

### Decision · Program_Chairs · 2021-01-07
**Final Decision**

**Decision:**

Accept (Poster)

**Comment:**

There is a consensus among the reviewers that the work is interesting and the paper should be accepted.  Nevertheless, several reviewers struggled with understanding the details. While the authors  (largely successfully) addressed these concerns, I believe that the paper is still too dense and hard to follow, I would encourage the authors to invest more time into improving its readability.  One important point which came late in the discussion is the provenance of baseline scores in the result tables (see the review by AnonReviewer3, the current manuscript claims that the numbers are taken from the original papers while in some cases, the numbers cannot be located in these papers). Unfortunately, the authors did not have a chance to respond to this criticism, and fortunately we could trace the key numbers and establish that the results are strong enough to warrant accepting the submission. Still, we would ask the reviewers to fix this issue in the final version.